# On the use of IASI spectrally resolved radiances to test the EC-Earth climate model (v3.3.3) in clear-sky conditions

Stefano Della Fera[1,2], Federico Fabiano[3], Piera Raspollini[2], Marco Ridolfi[4], Ugo Cortesi[2], Flavio Barbara[2], and Jost von Hardenberg[5,6]

[1]Department of Physics and Astronomy, University of Bologna, Bologna, Italy
[2]Institute of Applied Physics, National Research Council (IFAC-CNR), Sesto Fiorentino (FI), Italy
[3]Institute of Atmospheric Sciences and Climate, National Research Council (ISAC-CNR) Bologna, Italy
[4]National Institute of Optics, National Research Council (INO-CNR), Sesto Fiorentino (FI), Italy
[5]Department of Environment, Land and Infrastructure Engineering, Politecnico di Torino, Torino, Italy
[6]Institute of Atmospheric Sciences and Climate, National Research Council (ISAC-CNR), Torino, Italy

**Correspondence:** Stefano Della Fera (stefano.dellafera@unibo.it)

**Abstract.**

The long-term comparison between simulated and observed spectrally resolved Outgoing Longwave Radiation (OLR) can represent a stringent test for the direct verification and improvement of General Circulation Models (GCMs), which are regularly tuned by adjusting parameters related to sub-grid processes not explicitly represented in the model, to constrain the integrated OLR energy fluxes to observed values. However, a good agreement between simulated and observed integrated OLR fluxes may be obtained from the cancellation of opposite-in-sign systematic errors, localized in specific spectral ranges.

From the mid-2000s, stable hyperspectral observations of the Mid-Infrared region (667 to 2750 $cm^{-1}$) of the Earth emission spectrum have been provided by different sensors (e.g., AIRS, IASI and CrIS). Furthermore, the FORUM mission, selected to be the ninth ESA Earth Explorer, will measure, starting from 2027, the terrestrial radiation emitted to space at the top of the atmosphere (TOA) from 100 to 1600 $cm^{-1}$ filling the observational gap in the Far-Infrared (FIR) region, from 100 to 667 $cm^{-1}$.

In this work, in anticipation of FORUM measurements, we compare IASI Metop-A observations to radiances simulated on the basis of the atmospheric fields predicted by the EC-Earth GCM (version 3.3.3) in clear-sky conditions. To simulate spectra based on the atmospheric and surface state provided by the climate model, the radiative transfer model $\sigma$-IASI has been integrated in the Cloud Feedback Model Intercomparison Project (COSP) package. Therefore, on-line simulations, provided by EC-Earth model equipped with the new COSP + $\sigma$-IASI module, have been performed in clear-sky conditions with prescribed sea surface temperature and sea-ice concentration, every 6 hours, over a timeframe consistent with the availability of IASI data.

Systematic comparisons between observed and simulated Brightness Temperature (BT) have been performed in 10 $cm^{-1}$ spectral intervals, on a global scale over the ocean, with a specific focus on the latitudinal belt between 30°S and 30°N. The analysis has shown a warm BT bias of about 3.5 K in the core of the $CO_2$ absorption band and a cold BT bias of approximately 1 K in the wing of the $CO_2$ band, due respectively to a positive temperature bias in the stratosphere and a negative temperature bias in the middle troposphere of the climate model. Finally, considering a warm BT bias in the roto-

vibrational water vapor band, we have highlighted a dry bias of the water vapor concentration in the upper troposphere of the model.

## 25 1 Introduction

The Outgoing Longwave Radiation (OLR) flux, defined as the radiance emitted at top of the atmosphere integrated over the solid angle and over the spectral range from 100 to about 3330 $\mathrm{cm}^{-1}$ (3 - 100 $\mu$m), is a key quantity controlling the Earth Radiation Budget, and its accurate representation in General Circulation Models (GCMs) is crucial to get reliable historical and future simulations. For this purpose, GCMs are regularly tuned by adjusting parameters related to sub-grid processes not

explicitly represented in the model, to constrain the simulated OLR fluxes to observed values (Mauritsen et al., 2012; Hourdin et al., 2017), mainly provided by the Earth Radiation Budget Experiment (ERBE) (Barkstrom, 1984) and the more recent Cloud and Earth Radiant Energy System (CERES) mission (Loeb et al., 2018).

The availability of long-term measurements of radiative fluxes, extending over almost 40 years, makes them fundamental to assess the performance of GCMs. In this framework, Wild (2020) recently examined the radiative global budget of 40

state-of-the-art global climate models participating in the Coupled Model Intercomparison Project phase 6 (CMIP6) through a systematic comparison of broadband energy fluxes at surface and TOA with the CERES-EBAF dataset. The study has shown an important improvement of the CMIP6 models compared to the earlier model generations, but also a persistent inter-model spread, with a standard deviation of 2.8 $\mathrm{Wm}^{-2}$ for the all-sky OLR and of 2.6 $\mathrm{Wm}^{-2}$ for OLR in clear sky conditions.

Despite the comparison of observed and simulated broadband fluxes provides fundamental information about the perfor-

mance of climate models, the detection of model biases is complicated by the spectral integration, which may mask compensation errors in the OLR estimation. Conversely, spectrally resolved OLR contains the signatures of greenhouse gases (GHG), water and clouds and monitoring its behaviour by comparison to satellite measurements, offers an unprecedented opportunity to identify biases in GCMs and attributing them to a specific portion of the spectrum (Kiehl and Trenberth, 1997) and, thus, to a specific variable.

The first measurements of spectrally resolved radiances from space date back to 1970s, but only starting from the 2000s long-term and stable hyperspectral observations became available with the key satellite missions of the Atmospheric Infrared Sounding (AIRS, 2002-present) (Le Marshall et al., 2006), the Infrared Atmospheric Sounder Interferometer (IASI, 2006-present) (Clerbaux et al., 2009) and the Cross-track Infrared Sounder (CrIS, 2011-present) (Brindley and Bantges, 2016). The large amount of data available from these sensors opened interesting perspectives for the intercomparisons of instrumental

measurements and long-term analysis. Whitburn et al. (2020) computed OLR spectral fluxes starting from IASI radiances, using precalculated angular distribution models (ADMs), and compared IASI OLR integrated fluxes to CERES and AIRS broadband OLR products (Huang et al., 2008). Susskind et al. (2012) investigated the OLR interannual variability using AIRS data from 2002 to 2011 and compared the energy fluxes computed from spectrally resolved radiances to CERES broadband fluxes. In the same framework, Brindley et al. (2015) explored the interannual variability of spectrally resolved radiances at

different spatial scales by exploiting 5 years of IASI/Metop-A data. While the aforementioned instruments are able to provide

accurate measurements of the entire Mid-Infrared (MIR) portion of the spectrum, from 667 to 2500 $\mathrm{cm}^{-1}$ (4 - 15 $\mu$m), the Far-Infrared (FIR) spectral range, from 100 to 667 $\mathrm{cm}^{-1}$ (15 - 100 $\mu$m), which accounts for at least half of the Earth's energy emitted to space (Harries et al., 2008), still lacks of systematic measurements from satellite because of the intrinsic difficulties of development of the proper FIR technology (Palchetti et al., 2020). Planned for launch in 2027, the Far-Infrared Outgoing Radiation and Monitoring (FORUM) mission will fill this observational gap. FORUM will fly in loose formation with the IASI new generation (IASI-NG) on the Metop-SG-1A satellite (Ridolfi et al., 2020) thus, for the first time from space, the two instruments will cover the whole Earth's emission spectrum.

In this work, the OLR radiances at TOA are exploited to inspect and evaluate the performance of the EC-Earth climate model. In particular, we describe how the comparison between simulated and observed spectrally resolved clear-sky radiances can provide detailed information on model biases in temperature and humidity at different atmospheric levels, representing an alternative and reliable way, in addition to retrieval products and reanalysis datasets, to verify climate models performance.

For the comparison, we use a climatology of IASI clear-sky radiances built from Level 1C data, over the period 2008-2016. In future, the same approach may be applied to FORUM measurements.

In order to simulate upwelling OLR radiances starting from the climate model atmosphere, the fast radiative transfer model (RTM) $\sigma$-IASI (Amato et al., 2002) has been implemented in the CFMIP Observation Simulator Package (COSP v.1.4.1) inside the EC-Earth GCM. On-line historical simulations with prescribed sea surface temperatures (SSTs) and sea ice concentration (SIC) have been performed using COSP + $\sigma$-IASI in clear-sky conditions, in the MIR and FIR spectral regions, over the period 2008 - 2016, compatible with IASI available observations.

Using a similar approach, an existing negative bias in the OLR flux of about 4 $\mathrm{Wm}^{-2}$ in the AM2 GCM (Team et al., 2004), was investigated by Huang et al. (2006) by comparison of AIRS spectra to simulated radiances, and attributed to a water vapor transport deficiency of the model. In the same way, Huang et al. (2007) highlighted the existence of opposite-in-sign biases in water vapor and in $CO_2$ spectral bands, which produce fortuitous cancellations of spectral errors in the computation of the total broadband fluxes in the AM2 GCM.

In this work, the comparison between observed and model spectral radiance climatologies is preferred over the comparison between the climatology of atmospheric profiles (from the model and retrieved by IASI measurements) for the following reasons. The retrieval of vertical profiles from measured upwelling spectral radiances is a strongly ill-conditioned inverse problem, therefore a priori profile estimates are always used to constrain the retrieval. The used a priori information causes both global biases and local systematic smoothing errors in the retrieved profiles (Rodgers, 2000), thus making tricky the comparison of climatologies of profiles derived from the model and from the inversion of spectral radiance measurements (Rodgers and Connor, 2003).

The paper is organized as follows. In Section 2, models and observations are presented and briefly described. In Section 2.3, we introduce the implementation of the RTM in the COSP package in EC-Earth climate model. In Section 3 we present the results obtained by the long-term comparison between EC-Earth and IASI, we discuss the analysis method by highlighting the limits and the difficulties of the model-observations comparison in clear-sky conditions and draw the conclusions. Finally, in Appendix A, we recall the radiometric quantities used in the analysis.

## 2 Data and Methods

### 2.1 Models

#### 2.1.1 EC-Earth Climate Model

The EC-Earth climate model version 3.3.3 (Hazeleger et al. (2010), Döscher et al. (2021), http://www.ec-earth.org) is a state-
of-the-art, high-resolution Earth-system model participating in the last intercomparison project (CMIP6) (Eyring et al., 2016).
EC-Earth includes advanced, robust and validated components for the atmosphere (the Integrated Forecast System (IFS) model
cy36r4), the ocean (NEMO 3.6, (Madec et al., 2017)), sea ice (LIM3, (Fichefet and Maqueda, 1997)) and land processes (H-
Tessel, (Balsamo et al., 2009). The model has been tuned by minimizing the differences of radiative fluxes at TOA and at the
surface with respect to the observed fluxes from the CERES-EBAF-Ed4.0 dataset (Döscher et al., 2021).

In this work, atmosphere-only historical simulations have been performed with prescribed SSTs and SIC, from January
2008 to December 2016. The prescribed SSTs and SIC come from the AMIP protocol configuration for CMIP6 (Eyring et al.,
2016) and are provided as standard input to all models participating to CMIP6 (see also https://pcmdi.llnl.gov/mips/amip/ and
https://esgf-node.llnl.gov/projects/input4mips/). The dataset is created with the procedure described in Hurrell et al. (2008) and
merges the HadISST observational dataset (since 1870) to the more recent NOAA-OI (since 1981). EC-Earth reads the SSTs
and SIC as mid-month boundary conditions, which are then interpolated daily in the model run.

Furthermore, GHGs concentrations used in the simulation are derived from the standard dataset applied for the historical
CMIP6 runs (Meinshausen et al., 2017). More in detail, for the last 2 years (2015-16), we adopted the SSP2-4.5 scenario data,
which however matches observations until 2017 (Meinshausen et al., 2020).

In this configuration, the standard CMIP6 resolution TL255L91-ORCA1 is used; therefore, the atmospheric model IFS is
characterized by a horizontal resolution of approximately 80 km and by 91 vertical layers (Döscher et al., 2021).

In order to extract spectrally resolved OLR radiances from EC-Earth, we implemented the $\sigma$-IASI radiative transfer model
(RTM) (Amato et al., 2002) inside the COSP module (v 1.4.1), a simulator package able to map the climate model state into
synthetic observations which are directly comparable to the measurements of the real instruments (Bodas-Salcedo et al., 2011).
The current version of COSP implemented in EC-Earth includes simulators for passive sensors such as CLOUDSAT, MODIS
and MISR and active sensors like CALIPSO. It also provides a simulator of the International Satellite Cloud Climatology
Project (ISCCP) dataset and an interface for an old version (v. 9.1) of Radiative Transfer for Television and Infrared Observation
Satellite (TIROS) Operational Vertical Sounder (RTTOV), which can be linked to the package.

#### 2.1.2 The $\sigma$-IASI Radiative Transfer Model

$\sigma$-IASI is a monochromatic RTM able to simulate up-welling infrared radiances at high resolution (0.01 cm$^{-1}$), which can
be convolved with the Spectral Response Function (SRF) of any instrument. More specifically, it has been customized to
simulate the measurements by IASI-NG and of the future FORUM instrument. For each atmospheric layer, absorbing gas and
wavenumber in the 10 - 3000 cm$^{-1}$ range, the optical depths are computed using polynomial parametrizations determined on

the basis of accurate cross-sections computed by KLIMA, a validated line-by-line RTM developed at IFAC-CNR (Del Bianco et al., 2013; Cortesi et al., 2014). The inputs to the $\sigma$-IASI RTM are the surface pressure and temperature, the surface spectral emissivity, the profiles of temperature, humidity and concentrations of 11 gases ($O_3$, $CO_2$, $N_2O$, $CO$, $CH_4$, $SO_2$, $HNO_3$, $NH_3$, $OCS$, $HDO$, $CF_4$) and the cloud parameters (cloud cover, ice and liquid water content, effective radius of ice and liquid particles). The radiative transfer calculations are then performed using 61 fixed pressure levels and on a fixed wavenumber grid with a step of 0.01 $cm^{-1}$. The radiative code is also able to compute Jacobians with respect to all the geophysical variables, including the cloud parameters. The $\sigma$-IASI RTM has been extensively validated against IASI measurements (Liuzzi et al., 2017), Aircraft based Measurements (NAST-I) (Grieco et al., 2007) and ground-based measurements (Serio et al., 2008).

## 2.2 Observations

### 2.2.1 IASI

Part of the payload of the Metop series of EUMETSAT polar-orbiting meteorological satellites (Edwards and Pawlak, 2000), IASI is composed of a Fourier Transform Spectrometer and of an associated Integrated Imaging Subsystem (IIS), a broadband radiometer with a high spatial resolution for the co-registration with the Advanced Very-High-Resolution Radiometer (AVHRR) (Blumstein et al., 2004). Metop is characterized by a sun-synchronous orbit with equatorial crossing time at 9:30 AM (daytime) and PM (nightime) local times. IASI has been providing continuous data since October 2006, when it was firstly launched aboard Metop-A. It was followed by IASI-B (Metop-B), launched in 2012, and IASI-C (Metop-C), launched in 2018.

All the three IASI instruments cover the MIR spectral range, from 645 to 2760 $cm^{-1}$, with a spectral resolution of 0.5 $cm^{-1}$ and a spectral sampling of 0.25 $cm^{-1}$, for a total of 8461 spectral channels. The absolute calibration accuracy of the instrument is expected to be within 0.5 K ($https://iasi.cnes.fr/en/IASI/radiom\_res.htm$).

In order to obtain a uniform global coverage, IASI acquires measurements by scanning its Field of Regard (FOR) across the orbit track, with viewing angles that range from nadir up to 48.3 degrees on either side of the satellite track. Angularly, each FOR has a dimension of about 3.3° x 3.3° which, on ground, corresponds to a footprint of about 50 x 50 km at nadir. For each FOR (30 in total for scan), the instrument simultaneously acquires 4 spectra, each with a Field of View (FOV) of about 12 km of diameter at nadir.

In this work, we consider IASI data from the Fundamental Climate Data Record (FCDR) of reprocessed Metop-A Level 1c product ($DOI : 10.15770/EUM\_SEC\_CLM\_0014$), provided by EUMETSAT through the European Weather Cloud (EWC) service. On the basis of this dataset, which is homogeneous and validated over the whole selected time period (2008-2016), we build a monthly clear-sky radiance climatology on a global scale.

Firstly, we use the quality-flag (variable GQisFlagQual) available in the dataset to discard corrupted spectra. Then, among the 120 observed spectra of each scan across the satellite track, we only select those corresponding to the 8 pixels closest to the nadir view. The clear-sky spectra are detected by exploiting the cloud cover derived from the AVHRR (variable GEUMAVHRR1BCLDFRAC and GEUMAvhrr1BQual) (Guidard et al., 2011). In the same way, we distinguish the land/ocean

ground surface through the information (variable GEUMAvhrr1BLandFrac) provided by the AVHRR. More details about the IASI climatology will be provided in Section 3.2.

### 2.2.2   CERES

In this work, we exploit the CERES_SYN1deg_Ed4A products to get information about the observed cloud cover field on a global scale (Doelling et al., 2016). Among the various products, the dataset provides 1°-regional 3-hourly cloud coverage

derived from MODIS and geostationary satellites.

    The high temporal resolution of the product allows to easily find coincidences with IASI measurements and to analyze the large-scale atmospheric conditions where the IASI spectrum has been detected. As discussed in Section 3.2, this is useful for the analysis, since CERES data refer to an area (1° x 1°) of extension similar to the EC-Earth atmospheric resolution (0.7° x 0.7°). We also use the CERES Energy Budget and Filled (EBAF) dataset v.4.1 (Loeb et al., 2020) to estimate the observed

clear-sky broadband fluxes.

### 2.3   Implementation of the $\sigma$-IASI RTM in the EC-Earth climate model

We created a specific GCM-RTM interface inside the COSP module of the EC-Earth climate model in order to perform the radiative transfer calculations *online*, that is by passing instantaneous atmospheric fields on a global scale to the RTM with a time step of 6 hours.

In the radiative scheme of IFS, the spectral emissivity of the surface is assumed to be constantly equal to 0.99 outside the atmospheric window region (800 - 1250 $\mathrm{cm}^{-1}$). Conversely, within this region the emissivity depends on 8 types of surface: open sea, sea ice, interception layer, low and high vegetation, exposed and shaded snow and bare ground. These emissivity values are interpolated to a regular wavenumber grid with steps of 10 $\mathrm{cm}^{-1}$, in the range from 100 to 3000 $\mathrm{cm}^{-1}$ and supplied to the $\sigma$-IASI RTM. The surface pressure and surface temperature are directly supplied to $\sigma$-IASI, while the

simulated temperature, humidity and gases concentration profiles are first interpolated to the fixed pressure grid used by $\sigma$-IASI . Carbon dioxide, methane and nitrous oxide concentrations are horizontally and vertically uniform, depending only on time. The ozone mixing ratio used in the model simulation is a function of pressure, latitude and time, as described in Fortuin and Langematz (1995). Finally, concentrations of the other trace gases required by $\sigma$-IASI ($SO_2$, CO, $HNO_3$, $NH_3$, OCS, HDO, $CF_4$) are not modeled in the IFS, thus they are extracted from the U.S. Standard Atmosphere of the Atmospheric

Constituent Profiles dataset (Anderson et al., 1986).

    In order to minimize the huge impact of the radiative code on the GCM computing performance, the Look-Up Tables (LUTs) of optical depths parametrization coefficients are allocated and loaded from file only once at the beginning of the simulation, stored, and deallocated at the end of the process. Moreover, the outgoing radiance is computed only once every 4 latitude x longitude grid points of the EC-Earth model, for a total of about 6000 simulated spectra every 6 hours.

To limit the data storage required, the high resolution spectrum computed by $\sigma$-IASI is convolved with a 10 $\mathrm{cm}^{-1}$-wide box function and sampled every 10 $\mathrm{cm}^{-1}$. Since EC-Earth does not include variables with a spectral dimension, we stored the simulated spectra in new auxiliary 4D variables declared in the IFS grib code scheme, using the dimension corresponding

to vertical model levels as the spectral dimension. These simplifications allowed to strongly reduce the computational cost of the model run, passing from an initial value of 90000 core hours per simulated year (CHPSY) to 4000 CHPSY, which is comparable to the cost of the other simulators already present in COSP and about 8 times higher than an EC-Earth standard atmosphere-only simulation without COSP (about 500 CHPSY).

## 3 Results and Discussion

### 3.1 Sensitivity of a simulated OLR spectrum to atmospheric temperature and gas concentrations

In order to better correlate the differences between modelled and observed radiances to model biases, we first studied, for a reference tropical atmosphere, the sensitivity of the radiance, computed with $\sigma$-IASI, to model temperature and trace species concentration.

Figure 1 shows a spectrum of the TOA spectrally resolved radiance simulated by $\sigma$-IASI in clear-sky conditions, in the spectral range between 50 and 2250 $\text{cm}^{-1}$ . The spectral intervals measured by IASI and FORUM are highlighted, together with the approximated spectral ranges of the atmospheric window regions and the main gas absorption bands, which are summarised in Table 1.

The FIR region (from 100 - 667 $\text{cm}^{-1}$) is dominated by the signature of the rotational band of water vapor (blue shade), whose study will be consolidated with the help of future FORUM measurements (Brindley and Harries, 1998). In anticipation of FORUM measurements, we focus here on part of the MIR region of the spectrum measured by IASI, specifically between 645 - 2250 $\text{cm}^{-1}$. In this region, the spectrum undergoes a strong absorption between 640 - 750 $\text{cm}^{-1}$ due to $CO_2$. In more detail, in the core of $CO_2$ band (660 - 670 $\text{cm}^{-1}$), the atmosphere appears opaque from space and the radiance reaching TOA is originated from the stratosphere. On the contrary, in the wings of the $CO_2$ band measured by IASI (700 - 750 $\text{cm}^{-1}$) , the effective emission level is located in the middle- to upper- troposphere. From 800 to 950 $\text{cm}^{-1}$ and from 1100 to 1250 $\text{cm}^{-1}$ (red shades), the atmosphere is almost transparent and the radiance reaching the TOA mainly originates from the surface or the atmospheric layers closest to the surface. Other strong absorption bands are located between 980 and 1080 $\text{cm}^{-1}$ (ozone, green shade), between 1200 and 1400 $\text{cm}^{-1}$ (methane, pink shade) and between 1250 and 1350 $\text{cm}^{-1}$ (nitrous oxide, grey shade). Finally, the roto-vibrational water vapor band, located between 1400 and 1850 $\text{cm}^{-1}$, is highlighted.

The radiance reaching TOA originates mainly from upper atmospheric layers in the spectral regions of strong absorption, while in more transparent regions it originates from the lower atmospheric layers (Whitburn et al., 2021). More accurate information on the atmospheric layers contributing to the observed OLR spectrum can be extracted from the analysis of the Jacobians, defined as the partial derivatives of radiance with respect to any atmospheric parameters. In Fig. 2, we show the Jacobians of the most relevant variables computed with the $\sigma$-IASI RTM for a tropical standard atmosphere over ocean at the IASI sampling of 0.25 $\text{cm}^{-1}$ from 10 to 2250 $\text{cm}^{-1}$. For a better readability of the graph, the Jacobian values shown are the absolute values and normalized to their maximum value for each quantity, separately.

As we can see from Figure 2, the entire spectrum is sensitive to the temperature profile (red areas):

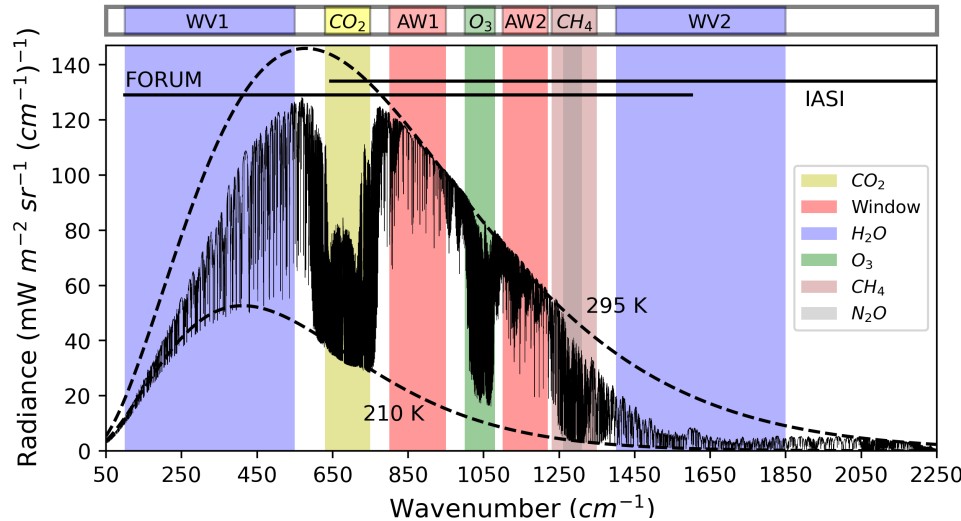

**Figure 1.** Spectrum simulated in clear-sky conditions over tropical ocean by the $\sigma$-IASI RTM. The main absorption bands are highlighted. Dashed lines show the equivalent blackbody emission at typical surface (295 K) and tropopause (210 K) temperatures.

| Acronym | Band Details | Spectral Range (cm$^{-1}$) |
|---------|--------------|----------------------------|
| WV1 | Water vapor (1) | 100 - 500 |
| CO$_2$ | Carbon Dioxide | 640 - 750 |
| AW1 | Atmospheric Window (1) | 800 - 950 |
| O$_3$ | Ozone | 980 - 1080 |
| AW2 | Atmospheric Window (2) | 1100 - 1250 |
| CH$_4$ | Methane | 1200 - 1400 |
| N$_2$O | Nitrous Oxide | 1250 - 1350 |
| WV2 | Water vapor (2) | 1400 - 1850 |

**Table 1.** Approximated spectral intervals of the atmospheric windows and the main absorption bands highlighted in Fig. 1.

– the atmospheric window *AW1* is more transparent than the atmospheric window *AW2*, where the radiation is slightly sensitive to the water vapor concentration. In the first case (*AW1*), the radiation is affected by the temperature of atmospheric layers between 0 and 3 km while, in the second one (*AW2*), it is controlled by the temperature of layers at greater heights, up to about 7 km;

– the FIR is strongly affected by the temperature of lower and medium troposphere [3 - 10 km];

– the CO$_2$ absorption band is mainly sensitive to stratospheric temperature [25 - 40 km] in the core of the band and to mid-to-upper tropospheric temperature [5 - 20 km] in the wing of the band;

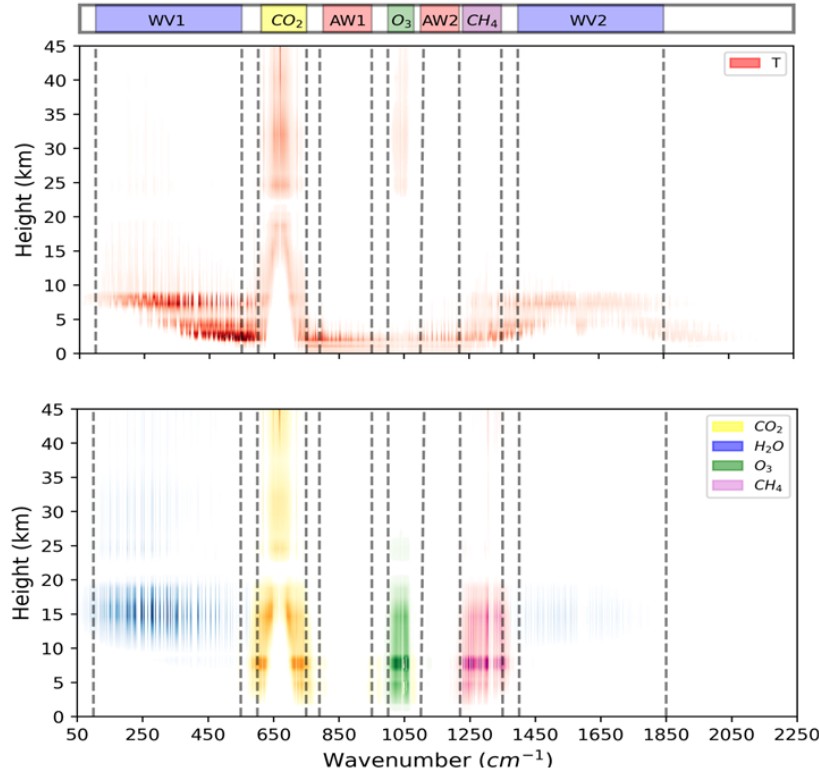

**Figure 2.** Absolute values of normalized Jacobians computed with $\sigma$-IASI for a tropical standard atmosphere with respect to the temperature (top panel) and gases concentration (bottom panel). $N_2O$ Jacobian is not shown because $CH_4$ Jacobian completely overlaps it.

- the $O_3$ band is affected by surface, lower troposphere and lower stratosphere temperature;

- the roto-vibrational band of water vapor *WV2* is sensitive to tropospheric temperature [3 - 10 km].

Specific features can be noticed for each gas:

- the outgoing radiance between 1400 and 1850 $cm^{-1}$ (*WV2*) is attenuated by $H_2O$ in the upper troposphere (blue area) from approximately 10 to 20 km. Water vapor also reduces OLR in the FIR region (*WV1*);

- between 980 and 1080 $cm^{-1}$, the ozone concentration strongly influences the spectrum over most of the troposphere (green area);

- at the same levels, the spectrum is affected by $CH_4$ concentration between 1200 and 1400 $cm^{-1}$ and by $N_2O$ between 1250 and 1350 $cm^{-1}$ (not shown);

– in the $CO_2$ band, the spectrum is sensitive to $CO_2$ concentration at the same height where it is particularly sensitive to temperature profile. Thus, it is influenced by $CO_2$ concentration in the stratosphere in the core of the band while in the wings of the band it is sensitive to $CO_2$ present in the troposphere.

A change of sign of the Jacobian is observed for all gases between troposphere and stratosphere (see Fig. 1 of Supplementary). In fact, in the troposphere absorption processes dominate the emission ones (so increased concentration reduces the OLR), while the opposite happens in the stratosphere.

## 3.2 Model-Observation comparison strategy

As already mentioned, due to the complication in comparing all-sky measurements at different spatial resolutions, here we focus only on the clear-sky part, leaving the analysis of cloudy sky to a future work. In addition, the low temporal sampling of the model (the RTM is called every 6 hours) and the uncertainties in the land surface emissivity do not allow to perform accurate comparisons between measured and observed radiances over land. Therefore, we focus on the comparison of day-time measurements and model outputs over ocean.

The selection process of simulated and measured spectra used to build the clear-sky radiance climatologies goes through the following steps. To save computing time, ECE simulates spectra in correspondence of only once every 4 latitude x longitude grid cells (see Fig.2 of Supplementary). The dimension of model cells is $0.7 \times 0.7°$. For each of these cells, we compute the monthly average radiance using only the simulated spectra with local solar time between 6 and 12 hours over the ocean, only if the current cloud cover of the model cell is less than 30 %. Generally, the radiative computation in clear-sky conditions in climate models exploits the same all-sky properties (e.g., surface temperature, temperature/humidity profile, surface albedo, aerosol), but with clouds removed. Since the temperature and humidity profiles of the model are indirectly affected by the presence of clouds, this causes a systematic negative bias when comparing observed and simulated clear-sky radiative fluxes. According to Sohn et al. (2006), this difference can reach up to -12 $\mathrm{Wm^{-2}}$ in the convectively active regions at tropical latitudes. In the CERES-EBAF 4.1 dataset, a new adjustment factor is introduced to generate TOA clear-sky fluxes that are more in line with the clear-sky fluxes represented in climate models, as described in Loeb et al. (2020) and in Loeb et al. (2018). On the contrary, in the EC-Earth vs IASI comparison, to mitigate this problem, we selected only the spectra computed over geophycal grid points where the simulated total cloud cover is less than 30%. This threshold is the result of a trade-off between reducing the impact of this potential source of bias and keeping a significant number of measurements in the analysis. In principle, a lower threshold would be more desirable, but at the same time this reduces the statistics. We then compute the monthly zonal averages by averaging the monthly mean radiances relating to the model cells within the considered latitude belt. With this procedure all model cells contribute to the zonal mean with equal weight.

Concerning the measurements, IASI spectra are selected from $2° \times 2°$ cells centered on the ECE model cells for which spectra are simulated. On the one hand, the dimension of these cells is large enough to allow the selection of a sufficiently large number of IASI spectra. On the other hand, these cells do not overlap each other, thus each IASI measurement contributes only once

to the statistics. For each of these cells, we compute the monthly average radiance using IASI measured spectra that meet the following conditions:

– The radiance is measured in day-time, in the near-nadir geometry, over the ocean, and corresponds to clear-sky conditions (cloud mask of AVHRR = 0).

       – The measured radiance falls into a CERES grid cell, measured within 3 hours from the IASI observation time, with cloud cover less than 30 %. Since CERES grid cells have a dimension of 1x1 degree, similar to the ECE model cells, applying the same threshold to the cloud cover we ensure consistency of the atmospheric conditions between model and
observations.

Finally, we compute the monthly zonal averages of observed radiances by averaging the monthly means obtained at the 2°x2° cells falling within the selected latitude belt.

Since we have few measured spectra over polar regions and the filter used to select clear-sky scenes is less accurate here, we limit the comparison for the latitudes ranging from 60°S to 60°N. The number of selected spectra is not homogeneously
distributed and most of the selected spectra are located in the subtropics ([15°-30° N] and [15°-30° S]), corresponding to the descending branch of the Hadley Cell. Moreover, the used filters particularly affect the mid-latitudes ([45°-60° N] and [45°-60° S]), where only few IASI pixels survive to the selection process. This is one of the reasons why we will mainly focus on the tropical regions ([30°S-30°N]), where we have a large number of both modeled and observed spectra (see Figures 3 and 4 and Table 1 of Supplementary).

## 3.3    Assessment of EC-Earth spectral biases in simulated clear-sky radiances with respect to IASI measurements

On the basis of these assumptions, a systematic comparison has been performed using a dataset that covers the years from 2008 to 2016, for latitudes ranging from 60°S to 60°N, over the ocean.

Figure 3 shows the 9 years zonal average of Brightness Temperature (BT) (defined in Appendix A) differences (model minus observations). Plots describing BT biases allow us to show more clearly the differences over all the selected spectral range.
On the contrary, radiance biases in the Plankian tail tend to become too small to be visible. Considering that model SSTs are constrained to be equal to the observed values, we expect small differences between model and IASI spectral radiances in the atmospheric spectral windows (*AW1, AW2*). Thus, the limited discrepancies in BTs obtained in the spectral window *AW1* in the tropical belt [30° S, 30° N] confirm the self-consistency of the performed comparison. Instead, as mentioned in Sec. 3.1, the atmospheric window *AW2* is more sensitive to the presence of water vapor and shows a small positive bias, approximately
equal to 0.3 K. At mid latitudes, however, especially in the Southern Hemisphere, a negative model bias of about 1 K is present in both the atmospheric windows, thus making difficult the comparison at all the frequencies at these latitudes. As mentioned before, this model bias is thought to be linked to the limited number of spectra available at these latitudes and also to the cloud cover representation in the model. This aspect will be be further discussed in Sect. 3.4.

Significant discrepancies, of about 3.5 K, are present in the $CO_2$ band at all latitudes, which might indicate a warm bias in
the model temperature of the upper-troposphere and stratosphere. A warm bias of about 1 K is also seen in the roto-vibrational

water vapor band (*WV2*) but this is limited to the tropical latitudinal belt between 30°S and 30°N. In a similar way, the bias visible in the $O_3$ band is strictly dependent on latitude and is characterised by a positive sign at the tropics while it tends to take negative values at mid latitudes. As described in Sect. 3.1, this spectral band is affected by surface, lower troposphere and stratospheric temperatures.

On the basis of the above considerations, we focus our analysis on the discrepancies found over tropical ocean where the BT differences in the atmospheric windows are close to zero.

Figure 4 shows the 2008-2016 average of model simulated and observed BTs over ocean, at tropical latitudes [30° S, 30°N] (top panels) and their differences (bottom panels). In this case we see that the model is generally in good agreement with the observations and the most significant discrepancies are found in the $CO_2$ band, in the $O_3$ band and along the water vapor

absorption band (*WV2*).

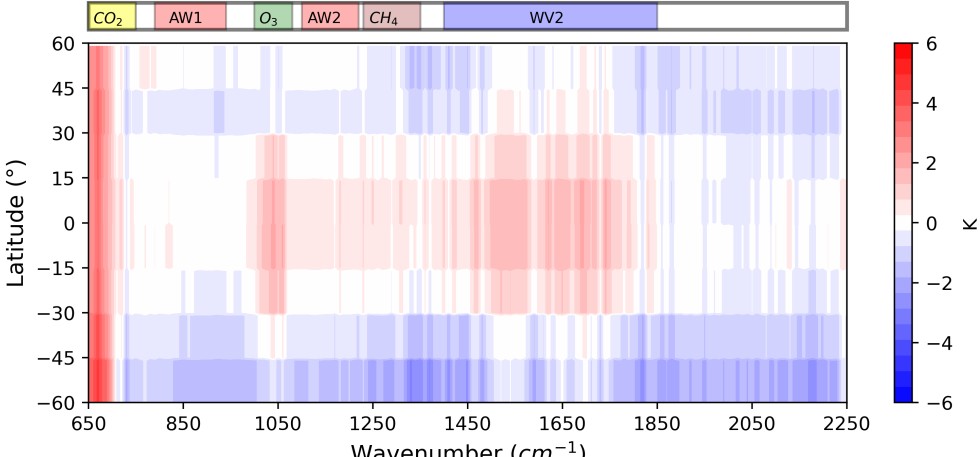

**Figure 3.** Brightness Temperature (BT) differences (model - observations) over ocean, averaged over the period 2008 - 2016

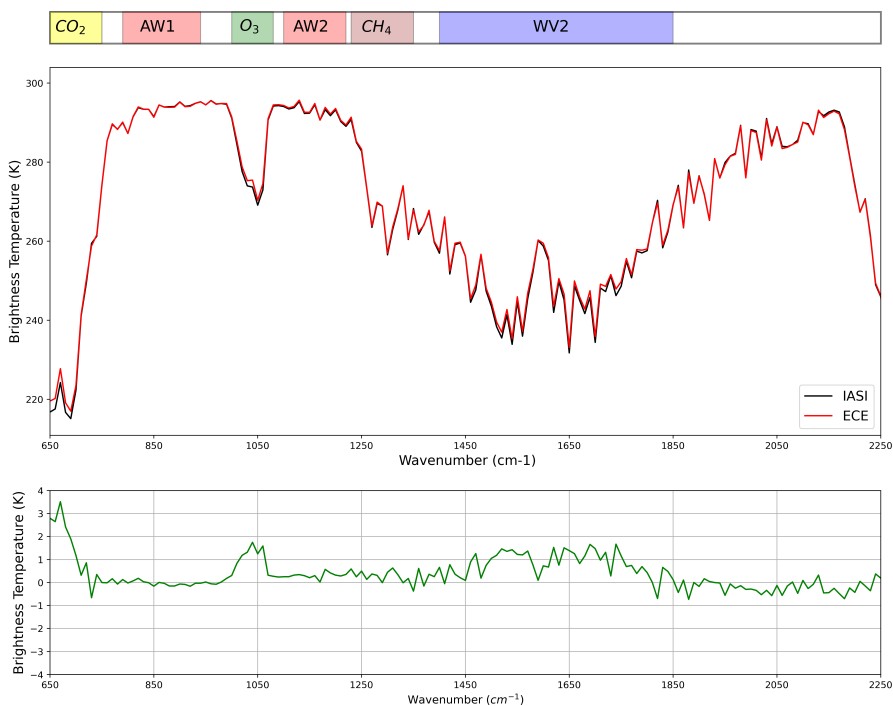

**Figure 4.** Average (2008-2016) Brightness Temperatures computed by EC-Earth and measured by IASI over the tropical ocean [30° S, 30°N] (top panel). The bottom panel shows the BT differences model minus observation.

It is possible to characterise the height dependence of the model temperature bias by focusing the analysis on specific spectral bands. Therefore, Figure 5 shows the 9-years monthly averages of simulated and observed BTs over ocean in four spectral intervals together with the respective temperature Jacobians, which highlight the atmospheric levels that are most sensitive to the temperature.

The largest sensitivity to temperature in the stratosphere is found in the spectral interval centered at 660 $\mathrm{cm}^{-1}$ (Panel A), while the channel at 700 $\mathrm{cm}^{-1}$ is sensitive to temperature in the upper troposphere and stratosphere (Panel B). The maximum sensitivity to temperature in the mid-troposphere is reached in the spectral interval centered at 730 $\mathrm{cm}^{-1}$. Finally, as usual, the spectral channels in the atmospheric window, in this case averaged between 845 and 855 $\mathrm{cm}^{-1}$, are a proxy of the lower troposphere and surface temperature. As already mentioned, the spectral intervals centered at 660 $\mathrm{cm}^{-1}$, 700 $\mathrm{cm}^{-1}$ and 320   730 $\mathrm{cm}^{-1}$ are not only sensitive to temperature but also to $CO_2$ concentration. The model, however, uses $CO_2$ global average concentrations smoothly increasing with time according to the actual measurements, thus any uniform warm model bias cannot be attributed to an erroneous carbon dioxide concentrations (see Section 2.1). The regional and seasonal variabilities of $CO_2$ concentrations amount at most to a few ppm, thus could cause only small local, seasonal biases.

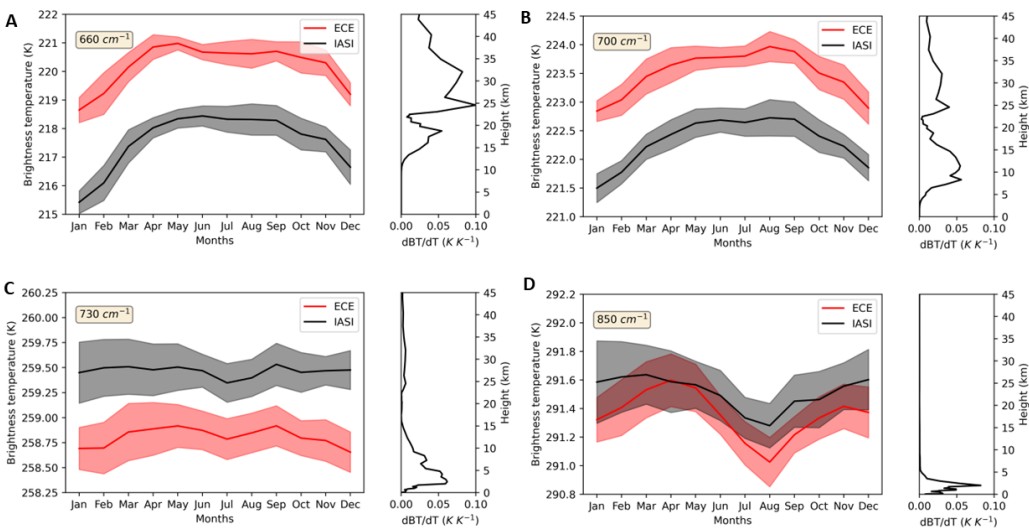

**Figure 5.** Brightness Temperature (BT) averaged in different spectral intervals over tropical ocean [30° S, 30°N] and their respective Jacobians of BT with respect to the temperature. The red line identifies the model BT while the black line describes the observed BT. The shadow areas represent the standard deviations. Note the different vertical scales used in the plots.

Therefore, Panel A of Figure 5 confirms the presence of a strong stratospheric warm bias in the model. The more we move towards the lower layers of the atmosphere (panels A and B of Fig. 5), the more the bias is reduced, until its sign reverses in the spectral band centered at 730 cm$^{-1}$ (see panel C), which is sensitive to the mid-tropospheric temperature. Finally, as expected, the bias is very small in the atmospheric window over ocean (Panel D). A more peaked seasonality is present in the ECE curve, which is however within the standard deviation of the two curves. BT differences between ECE and IASI in the spectral intervals of Figure 5 are also shown on a global scale in Figures 5,6,7,8 of the Supplementary. From these plots it is evident that the biases are homogeneous over the tropical and sub-tropical latitudes, where we are comparing the simulated and observed BT. Some compensating biases are only present at 850 cm$^{-1}$, in the atmospheric window. However, these differences are generally very small, always within 1 K.

We now exploit the intervals 725-735 cm$^{-1}$ and 1395-1405 cm$^{-1}$ to explore the accuracy of the representation of the water vapor concentration in the model. In fact, as illustrated by the integrated Jacobians reported on the left panel of Fig. 6, the spectral band at 1400 cm$^{-1}$ (*WV2*) is sensitive both to the tropospheric temperature and to the upper tropospheric water vapor concentration. In both spectral intervals, the maximum sensitivity to temperature occurs between 3 and 10 km (green and pink lines on the left panel of Fig. 6). Since the previous analysis (panel C of Fig. 5) has shown a small negative BT bias at 730 cm$^{-1}$ assigned to a cold bias of the mid-tropospheric temperature in the model, if the water vapor concentration were well represented, we would see a negative BT bias in the spectral band 1395-1405 cm$^{-1}$. However, since the model BT in the spectral interval centered at 1400 cm$^{-1}$ shows a slightly positive bias (Figure 6, right panel), we conclude that the negative temperature bias of the model seems to be over-compensated by a dry bias of the water vapor profile in the 7 - 15 km

range. In fact, a too dry upper troposphere in the model allows more radiant energy to reach the TOA, as also witnessed by the negative sign of the water Jacobian shown on the left panel of Fig. 6. BT differences between ECE and IASI in the spectral band 1395-1405 $cm^{-1}$ are also shown on a global scale in Figure 9 of the Supplementary. Also in this case, the positive bias is fairly uniform in the latitudinal band between 30°S and 30°N.

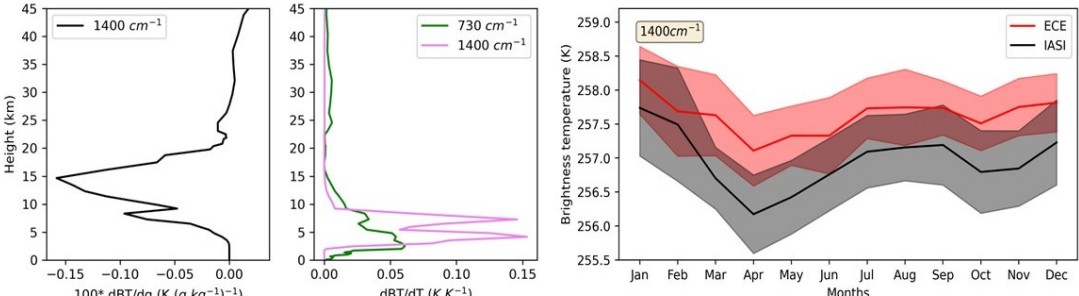

**Figure 6.** Left: Jacobian of water vapor integrated over the spectral band 1395 - 1405 $cm^{-1}$ and Jacobian of the temperature integrated over the spectral band 1395 - 1405 $cm^{-1}$ and 725-735 $cm^{-1}$. Right: Comparison of Brightness Temperature in the spectral band centered at 1400 $cm^{-1}$.

Finally, as in the $CO_2$ case, EC-Earth uses local climatological monthly means of methane and ozone concentrations, thus, the discrepancies occurring in the ozone and methane absorption bands (not shown) most likely are due to biases of the simulated temperature.

For the same period, the average clear-sky OLR flux computed by EC-Earth over ocean, between the latitudes 30°S and 30°N is equal to $288.47 \pm 0.34$ $\text{Wm}^{-2}$. This is slightly overestimated compared to the analogous average clear-sky flux obtained from CERES observations, that is equal to $287.36 \pm 0.32$ $\text{Wm}^{-2}$. From the Stefan-Boltzmann law, considering the power radiated by a black body at the temperature of $295$ $K$ (about the average surface temperature of tropical ocean), a difference of $1$ $\text{Wm}^{-2}$ corresponds to a BT difference of about $0.2$ $K$, i.e. smaller than the biases localized in specific wavenumber ranges that we found from the spectral analysis.

To date, systematic FIR spectral radiance measurements from space are not yet available, thus we are not able to characterize the discrepancies between model and observations in the whole OLR spectral range. Despite of that, the analysis presented shows clearly that a good agreement between simulated and observed total OLR fluxes may be obtained from the cancellation of opposite-in-sign systematic errors, localized in specific spectral ranges. In conclusion, observations of spectrally resolved OLR fluxes from space are needed for a proper tuning of model parameters.

## 3.4 Discussion

We have seen that a perfect spatial and temporal matching of measurements and simulations is very difficult to actualise, therefore, systematic biases could also arise from the strategy adopted to sample the data. In order to evaluate the impact of the data sampling strategy, we carried out the following test. We interpolated the EC-Earth model cloud fraction and the measured CERES cloud fraction to a regular space grid of 1° x 1° and time step of 6 hours. Then, assuming alternatively the interpolated CERES and EC-Earth cloud fractions, we built the statistical distributions for the year 2008 of the same observed SST for the grid points with cloud fraction less than 30 %. Figure 7 shows the SST statistical distributions obtained for EC-Earth and CERES cloud fractions, at tropical- (left panel) and mid- (right panel) latitudes. At tropical latitudes the SST distributions obtained with the model (red boxes) and CERES (grey boxes) cloud fractions are quite similar: the average values differ by 0.4 K and the standard deviations (≈ 4.5 K) differ by less than 0.1 K. On the other hand, at Southern mid-latitudes (-60° S, -45° S, see the right panel of Fig. 7) the offset between the two distributions amounts to 0.9 K. The larger bias of 0.9 K at the Southern mid-latitudes is likely contributing to the observed negative model BT bias found in the atmospheric window at the Southern mid-latitudes in Figure 3. The good agreement between the two SST distributions found in the tropical latitude belt strengthens our confidence on the previous analyses we presented for tropical latitudes. At these latitudes, the choice of comparing model and measured climatologies corresponding to cloud fractions smaller than 30 % ensures that the biases introduced by the data sampling strategy is smaller than ≈0.5 K, i.e. also smaller than most of the model biases inferred from Fig. 5.

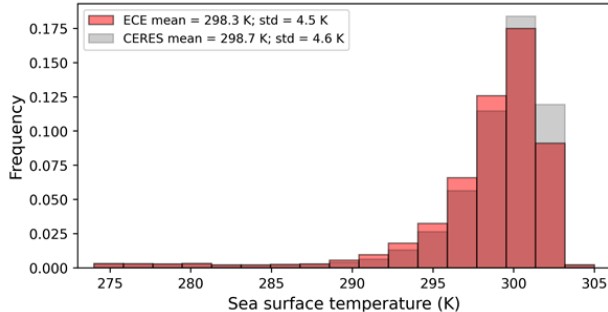 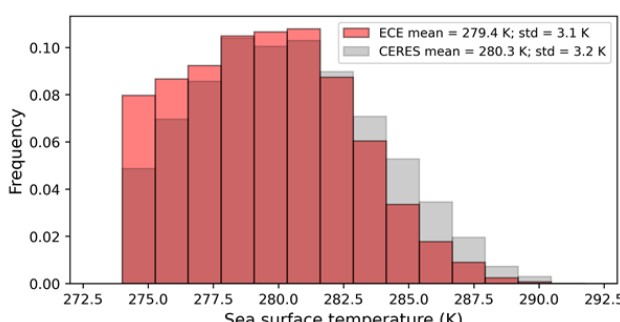

**Figure 7.** Distribution of sea surface temperature of the model for cloud fraction < 30 %, assuming the EC-Earth (red) or the CERES (grey) cloud fractions. The distributions were computed for tropical- [-30 °S, + 30° N] latitudes (left) and mid- [-60 °S, - 45° S] latitudes (right).

We further tested the results of the spectral analysis by comparing the temperature and humidity obtained from the climate model outputs with data provided by ERA5, the latest climate reanalysis from ECMWF. The reanalysis combines available data from different instruments (satellites, ships, weather stations etc.) with models, to generate a complete and continuous global coverage of the main geophysical variables (Hersbach et al., 2020). The left panel of Fig. 8 shows the temperature differences between EC-Earth and ERA5 reanalysis, averaged over 15 years (2000 - 2014). The strong warm bias in the stratosphere

confirms the discrepancy found in our spectral analysis in the region between 655 and 665 $cm^{-1}$. On the other hand, our spectral analysis did not directly detect the cold model bias visible in Fig. 8 at the tropopause. In fact, as shown in Fig. 2, the OLR spectrum is not very sensitive to temperature and gas concentrations at these heights. In addition, the spectral band centered at 700 $cm^{-1}$ (Panel B of Fig. 5) is partially affected by the positive stratospheric temperature bias, which can easily mask the underlying negative bias at the tropopause. We also note that the small negative model temperature bias present in the troposphere is consistent with the difference in BTs found in the spectral band at 730 $cm^{-1}$ (panel C of Fig. 5).

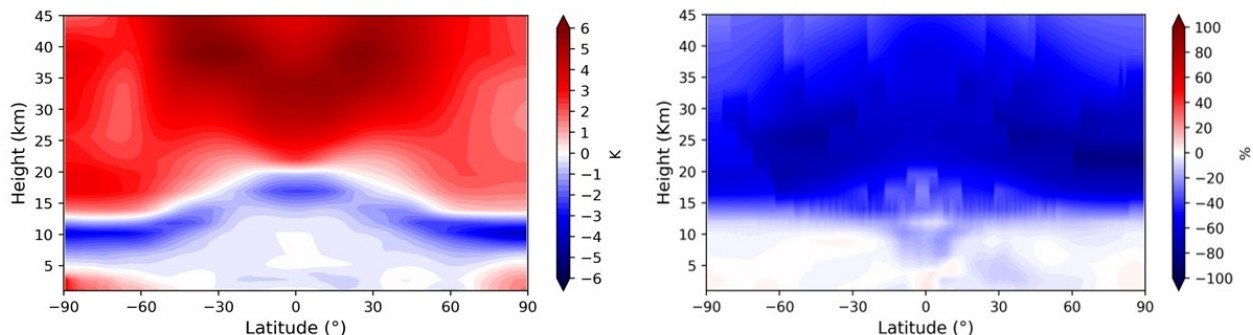

**Figure 8.** Differences betweeen EC-Earth climate model and ERA5 reanalysis temperatures (left) and percentage differences between EC-Earth climate model and ERA5 specific humidity. Differences are a 15 years [2000 - 2014] average.

The right panel of Fig. 8 shows the percentage differences between EC-Earth and ERA5 specific humidity, averaged over the period 2000-2014. The absolute values of these differences are visible in Figure 10 of Supplementary. The negative EC-Earth humidity bias found in the upper troposphere and stratosphere is consistent with the positive bias detected on the model simulated BTs in the spectral interval between 1395 and 1405 $cm^{-1}$. Indeed, the drier atmosphere of the model at these heights implies a greater amount of radiation reaching the TOA in the water vapor band.

We also provide an estimation of the BT bias inferred from the temperature and water vapour biases of the EC-Earth model with respect to ERA5. First, we computed the mean profile of these temperature and water vapor biases along the tropical latitudes between 30°S and 30°N. Then, we performed the scalar product of these profiles with the respective Jacobians (Figures 5 and 6), computed assuming the profiles of a standard tropical atmosphere. The result of this estimation is summarized in Table 2.

The clear-sky BT biases found in the comparison with IASI (Section 3.3) are generally consistent with the estimates inferred here from the comparison with ERA5. Remarkably, the sign is consistent in all cases, although some differences are found in the magnitude. The spectral analysis indicates a stronger positive bias in the stratosphere (660 $cm^{-1}$) than inferred from ERA5. This is also seen in the spectral band at 700 $cm^{-1}$. In addition, we also have a more pronounced negative bias at 730 $cm^{-1}$, possibly produced by a larger negative temperature bias of the model in the middle troposphere. Finally, the BT biases for the

| Spectral interval center ($\text{cm}^{-1}$) | [660] | [700] | [730] | [850] | [1400] |
|---|---|---|---|---|---|
| Height region with temperature sensitivity (km) | [25-45] | [5-15] and [20-35] | [3-10] | [0-5] | [3-10] |
| Height region with water vapour sensitivity (km) | – | – | – | – | [5-20] |
| BT bias expected from ERA5 intercomp. (K) | +2.5 | +0.8 | -0.2 | -0.2 | +0.7 |
| BT bias obtained from IASI intercomp. (K) | +3.5 | +1.5 | -0.8 | -0.2 | +0.5 |

**Table 2.** BT biases expected from the comparison between profiles of climate model and ERA5, and observed in the comparison of climate model simulations and IASI observed climatologies.

spectral bands at 850 and 1400 $\text{cm}^{-1}$ show a very good agreement with the estimates obtained from the ERA5 reanalysis. The positive BT bias at 1400 $\text{cm}^{-1}$ is caused by the negative bias of the model water vapour in the upper troposphere.

It is important to note that the discrepancies between the BT biases, described in the last two rows of Table 2, may arise from different causes. First, the comparison of temperature and water vapor concentration profiles of Figure 8 are performed in all-sky conditions, while the spectral analysis reflects the differences observed in clear-sky. Furthermore, the Jacobian used in the estimation of Table 2 is referred to a standard tropical atmosphere, on the basis of the dataset of Anderson et al. (1986), which may differ from the real one.

## 4   Conclusion and future perspectives

In the measured spectral radiances, the signatures of the main climate variables can be identified, separated and used to assess climate model biases. This analysis can not be carried out on the basis of intercomparisons of total OLR fluxes that, as seen by the comparison of EC-Earth output and CERES observations, can easily hide compensating errors.

Thus, the availability of long-term measurements of spectrally resolved OLR radiances offers new important perspectives to strictly evaluate GCM performance. Indeed, the spectra measured on a global scale represent a more accurate benchmark than that provided by reanalysis datasets, which are computed using both observations and model simulations. In addition, the spectral analysis is not affected by the systematic biases affecting atmospheric profiles retrieved from remote sensing measurements by applying a priori constraints to otherwise severely ill-conditioned inversions.

We implemented the $\sigma$-IASI RTM in the COSP package in order to perform on-line simulations of synthetic clear-sky spectra starting from the EC-Earth GCM profiles on a global scale, with a time step of 6 hours for the period 2008-2016. Thus, we compared the EC-Earth-simulated spectral radiances to the IASI-measured radiances built from the Fundamental Climate Data Record (FCDR) of reprocessed Metop-A Level 1c product on a frequency grid of 10 $\text{cm}^{-1}$. We limited the analysis to the clear-sky conditions identified by grid-points where the observed (CERES) and simulated (EC-Earth) cloud fraction is smaller than 30%. We found that such a small threshold limits the indirect effect of clouds on radiation in the model. The comparison has been performed on a global scale ensuring the spatial and temporal coincidence between the modelled and observed spectra. We focused on the day-time tropical ocean [30°S, 30°N] area, where the analysis is not affected by the uncertainties due to the land emissivity and the discrepancies between observed and simulated radiances in the atmospheric window are close to zero.

The spectral analysis carried out in this conditions leads to the detection of the following EC-Earth model biases which, due to compensations, do not show up in the comparison of the total OLR fluxes:

– A strong ($\approx$3.5 K) positive temperature bias in the stratosphere.

       – A ($\approx$1 K) negative temperature bias in the troposphere.

       – A ($\approx$0.5 K) positive bias in the BTs in the water vapor band, indicating an under-estimation of the model water vapor in the upper troposphere.

Finally, we have shown that the results of our spectral analysis are generally consistent with those obtained by compar-
ing EC-Earth temperature and specific humidity profiles to the ERA5 reanalysis. The largest discrepancy between the two intercomparisons is found in the stratospheric temperature bias but the differences are still within 1 K.

The next phase of the work will extend the analysis also to spectral radiances in the presence of clouds, whose impact on the radiation represents the greatest source of uncertainty in climate models. The objective is to perform a spectral analysis of the cloud radiative effect and to inspect spectral model biases by comparing climate model outputs to observations in all-
sky conditions. Finally, the same approach could be extended to other climate models and, in the near future, it will involve FORUM FIR measurements for a comprehensive analysis of the climate model ability in reproducing the whole Earth emission spectrum.

*Code availability.*  Permission to access the EC-Earth3 source code can be requested from the EC-Earth community via the EC-Earth website (`http://www.ec-earth.org/`, EC-Earth consortium, 2019a) and may be granted if a corresponding software license agreement is
signed with ECMWF.

The $\sigma$-IASI code is available on Zenodo (`DOI:10.5281/zenodo.7019991`).

IASI data can be downloaded from EUMETSAT data center (`https://www.eumetsat.int/eumetsat-data-centre`).

Scripts and model data used for the analysis are available on Zenodo at `https://doi.org/10.5281/zenodo.6912765`

## Appendix A:  Brightness temperature and spectral OLR flux

The spectral radiance $L_\nu$ at wavenumber $\nu$, can be converted into Brightness Temperature (BT) by using the inverse of the Plank function. Specifically, brightness temperature is defined as the temperature $T_\nu$ of the black-body $B_\nu$ that emits the same radiance $L_\nu$ at wavenumber $\nu$. Thus we set:

$$L_\nu = B_\nu(T_\nu) = \frac{2h\nu^3 c^2}{e^{\frac{hc\nu}{kT_\nu}} - 1} \tag{A1}$$

where $h$ is the Planck's constant, $c$ is the speed of light in vacuum and $k$ the Boltzmann's constant. Inverting this formula we
get:

$$T_\nu = \frac{hc\nu}{k\ln(1 + \frac{2hc^2\nu^3}{L_\nu})}. \tag{A2}$$

The spectral radiance is the energy flowing through the unit area, per unit time, per unit wavenumber and solid angle. In general, the spectral radiance is not isotropic, it usually depends on the orientation of the considered solid angle. This orientation can be identified using the zenith and the azimuth angles $\theta$ and $\phi$, respectively. Thus, the spectral radiance, in general is a function $L_\nu(\theta, \phi)$. The spectral flux $F_\nu$ is defined as the integral of the radiance over an hemisphere of solid angle:

$$F_\nu = \int_\Omega L_\nu(\theta, \phi) \cos(\theta) d\Omega \tag{A3}$$

where $d\Omega$ is the infinitesimal element of the solid angle $\Omega$, the hemispheric domain of integration. In spherical coordinates we get $d\Omega = \sin(\theta) d\theta d\phi$. Thus, the spectral flux $F_\nu$ is written as:

$$F_\nu = \int_0^{2\pi} d\phi \int_0^{\pi/2} d\theta L_\nu(\theta, \phi) \cos(\theta) \sin(\theta) \tag{A4}$$

The total OLR flux is the integral of $F_\nu$ over the OLR spectral range, usually defined from 100 to 3333 cm$^{-1}$ (or from 3 to 100 μm in wavelength).

*Author contributions.* SDF interfaced the $\sigma$-IASI radiative transfer model in the COSP package with the help of FF, MR and JVH. SDF and FF configured the climate model simulation and post-processed climate model data with the support of JVH. SDF developed the code to create the climatology of IASI measurements, with the help of UC, PR, FB and MR. SDF drafted the manuscript, and all authors contributed to its final version.

*Competing interests.* The authors declare that they have no conflict of interest

*Acknowledgements.* The authors acknowledge CINECA for proving computational resources through the Italian SuperComputing Resource Allocation (ISCRA, projects ECECOSP and ECEIASI) and EUMETSAT for making available the huge amount of L1c IASI data through the European Weather Cloud service (EWC). The $\sigma$-IASI RTM was made available by its authors in the frame of the "FORUM-scienza" (FORUM science) project, agreement No. 2019-20-HH.0, funded by the Italian Space Agency in the 2019-2022 time frame.

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
