# Peer review of "On the use of IASI spectrally resolved radiances to test the EC-Earth climate model (v3.3.3) in clear-sky conditions"

_EGUsphere, 2022_

## Referee Comment (RC1)

The paper "On the use of IASI spectrally resolved radiances to test the EC-Earth climate model (v3.3.3) in clear-sky conditions" by Della Fera et al. explores the use of top-of-Atmosphere out going spectral radiance measurements to evaluate biases in climate models and linked them to key variables of the Earth system. The method is applied to the Earth-system model EC-Earth discussing biases in the temperature and water vapour content. The paper is generally well organised and written and I propose it for publication after the following points have been addressed by the authors.

Major comments:

The selection of the clear-sky areas to analyse goes through a series of steps and eventually a set of clear sky spectra are obtained from the dataset of simulated radiances. Would it be possible to show a map with the regional density of pixels surviving the selection, to help identify which regions are most contributing to the analysis? For example, are the pixels used to create figure 3 distributed homogeneously or with a higher density somewhere? Or the analysis is robust enough to any regional bias?

Jacobians in Figure 2 are shown as normalized to the relative maximum. I understand that this is to show qualitatively the spectral sensitivity of the radiance to various parameters, but would it make sense to also show the absolute values somehow (for example in Figure 6)? It would be interesting for example to estimate if the biases in temperature and water vapour concentration shown in figure 8 would indeed reasonably map into the observed radiance/brightness temperature biases discussed for figure 6 and figure 5c.

Figure 5 panel B shows that EC-Earth has a positive temperature bias in the region 5-15km in the 30S-30N area as seen in the 700-710 cm-1 spectral region. Yet in the comparison against ERA5 that area shows generally a negative bias up to 15km (positive biases are seen north and south of 30 degrees, above the tropopause). This seems in contradiction to what it is said at line 360 Ch. 3.4

In the conclusion I'd like to see a clearer discussion of the limitations of the presented method, namely the difficulty of making sure that a meaningful comparison between simulated and measured radiances can be obtained (this is briefly touched in chapter 3.4 regarding sampling biases). The analysis presented in this work does indeed show the advantage of having spectrally resolved OLR computed from the model fields overcoming some of the limitations of retrieval products, but on the other hand, assumptions and parametrisations are still needed to perform accurate radiative transfer computations. Restricting the analysis to clear-sky ocean areas circumvents the difficulties of a correct simulation of surface radiative properties or the representation of cloud microphysical characteristics whose uncertainties might create larger differences than the model biases one attempts to investigate. Can these shortcomings be somehow mitigated or is the method really only suited for a limited set of cases?

Minor comments:

Sect. 1, line 39: "and, thus, to a specific parameter": what is it meant here by parameter? An active model variable such as temperature or concentration of a particular atmospheric constituent or rather a model parameter governing a particular process/parametrisation in the model? I think that it is difficult that a single model parameter, if the latter is intended, would be able to explain a bias in a particular spectral region. Indeed, many of the parameters tuned in EC-Earth to achieve satisfactory radiative fluxes at TOA, affect cloud parametrisations. Please, clarify.

Sect. 1, line 43: "1970s, but only starting from the 2000s long term…."

Sect. 1, line 56: "In anticipation of FORUM measurements….". This is repeated few time in the document: does it mean here something along the lines of "To demonstrate the potential of the future FORUM measurements"?

Sect. 1, line 65: "MIR" has not been defined while FIR yes.

Sect 1, line 79: "paves the way for direct assimilation" Not clear here: why the implementation of an online simulator would help in that direction?

Sect. 2.1.1, line 101: CLOUDSAT is an active radar. ISCCP is not a sensor, if it means the International Satellite Cloud Climatology Project. References and full description of the acronym would help here, as done for RTTOV mentioned later, for example.

Sec 2.2.1, line 137: is there a reference for the AVHRR cloud mask?

Sec 2.2.2: I think there should be references to the CERES dataset used

Sec 3.1, line 197: "partial derivatives of radiance with respect to the most relevant atmospheric parameter". Perhaps a more general definition like "with respect to any relevant atmospheric variable. Here we show the most relevant…"

Sec 3.3, line 298: The example discussed in this section shows that spectral OLR are indeed useful to provide a complete view of model biases, but it also shows the difficulty to select meaningful area to compare with measurement, thus limiting somehow the possibility to map directly a bias to a parameter in a global model. Perhaps the sentence can be rephrased to reflect this limitation.

Sec 3.3, line 330: "witnessed by the negative sign of the water Jacobian" -> "as represented by the negative values of the water vapour Jacobian"

Figure 8: The representation of the differences in specific humidity as percentage tends to highlight the stratosphere due to the very low water vapour concentration there. Absolute differences could perhaps be better related to the absolute value of the Jacobian that could be shown e.g. in Figure 6?

Sec 4, line 392: The results are only shown for ocean areas, land biases are not really discussed in depth in the text. Is it worth it reporting it in the conclusions?

---

## Referee Comment (RC2)

The reviewed manuscript uses the spectrally resolved outgoing radiances from the IASI sounder to evaluate the output of the EC-Earth climate model. The identified biases are linked to the main variables of the Earth-atmosphere system. The scientific work is topical and the methodology is sound but, in my opinion, the paper lack of structure here and there and the presentation could be improved. The analysis of the results should be sometimes more detailed. To my point of view, it will be suitable for publication after addressing/clarifying the following comments.

**General comments**

I think it should be made more clear, before starting the analysis of the differences between IASI and the model, what can be all the causes of the differences (biases in modeled temperature or atmospheric concentrations, methodology: effect of the cloud filtering, time matching between IASI and model, etc.). Mentioning first all the potential reasons for the presence of biases would help greatly for the clarity of the manuscript.

**Subsection 2.1.1**: Could the authors describe a bit more how the EC Earth model works (maybe accompanied by a conceptual flowchart)? In particular, I am not sure to understand from where are all the prescribed parameters coming from. To me, it should be explained a bit more in the manuscript. Or at least, the authors should mention clearly in what paper an extensive description of the model can be found.

**3.2 Model comparison strategy**:

I think this section is not clear enough and should be partly rewritten.

- For instance, "The model was run with prescribed SSTs and SIE, […]" It is not clear to me where are these information coming from (see my previous comment).
- If I understood well, the authors built monthly climatology of IASI radiance (near-nadir). Only cloud free pixels are considered for the averaging. What is the grid size of these monthly climatology? 1°x1°? Then, for the comparison with the model, from these monthly climatology, the authors only consider the grid cells that show a monthly mean cloud cover (according to CERES) lower than 30%. Is that right?
- How was this 30% threshold chosen? Why not 40% or 20% for example? I think it should be better justified.
- A map showing the grid cells that are kept for the comparison would be nice.
- Also, in the IASI monthly mean radiances, how many measurements are present on average per month in each grid cell? If the number is too low (which might be the case for some of the pixels as only near-nadir observations are considered), the comparison with the model might be biased as a few number of observations is not representative of a whole month. This should be kept in mind for the analysis. It might be good to add a postfiltering to remove from the comparison the grid cells that contain too few observations.
- Lines 252-253: "In fact, monthly means of observed and simulated outgoing radiances are computed over the same spatial grid by associating IASI measurements to the nearest EC-Earth grid point.". I am not sure I got the idea here. If I understood well, for the time correspondence between the model simulations and the IASI observations, only the model output of the 6-12 AM and PM time are considered? Both are then averaged to obtain a "daily" mean (and same for IASI, ascending and descending nodes are averaged to get a daily mean?)? And this is the reason why you only considered sea measurements for the analysis (because of the very high diurnal cycle of the radiance over land)?

**3.3 Assessment of EC-Earth biases in simulated clear-sky radiances with respect to IASI measurements**:

The descriptions in this section should be more quantitative. For example, line 274 (but also elsewhere): "[…] a small positive bias". Of how much?

I think some additional plots could help for the interpretation of the results:

- Are the biases the same every year or is there differences from year to year? Maybe a time series showing the differences between the model and IASI (averaged on the different bands WV1, WV2, … for example) would be nice. I am also wondering if there is a difference in the biases between day and night.
- Another plot that could greatly help for the interpretation of the results and could be a nice addition for the paper is global maps of the difference between IASI and the model for a few selected channels (instead of a zonal average of the difference) (as done in Whitburn et al. 2021 for example). In particular, this would allow to highlight potential compensating biases between different regions.
- It would be nice to see Fig. 3 in W m$^{-2}$ to evaluate more easily the impact of the differences (maybe in appendix).

**Whitburn, S.,** Clarisse, L., Bouillon, M. *et al.* Trends in spectrally resolved outgoing longwave radiation from 10 years of satellite measurements. *npj Clim Atmos Sci* **4**, 48 (2021). https://doi.org/10.1038/s41612-021-00205-7

**Specific comments**

**Abstract**

Line 15: If I am not mistaken, I think it is "Metop" and "MetOp". To be checked and replace everywhere.

Line 18-19: "[…] while a cold bias occurs over land". The authors didn't mention that they were talking about sea just before. Please clarify.

**Introduction**

Line 41: Missing references for AIRS and IASI.

Line 46: Also mention the existence of a spectrally resolved OLR product at 10 cm$^{-1}$ derived from AIRS measurements (e.g. Huang et al. 2008, Chen et al. 2016).
https://disc.gsfc.nasa.gov/datasets/AIRSIL3MSOLR_6.1/summary?keywords=AIRSIL3MSOLR_6.1

Appendix A: I don't really see the added value of Appendix A for the paper. The information presented are very general and the different formula can be found in many atmospheric science books and published papers.

**Data and Methods**

Line 91: Please defined IFS.

Line 132: FCDR: missing citation. Also, mention that this dataset is reprocessed with the latest version of the L1C and is consistent over the whole time series. I guess this is why you chose to use it?

Line 136: […] the 8 pixels closest to the nadir view. Thus, the clear-sky […]. There is a problem with the transition between the two sentences here.

Line 137: AVHRR cloud dataset: cite Guidard et al. (2011). Is there a reason why this cloud product was chosen over the L2 cloud product? For consistency over the whole time series? Maybe it would be nice to justify it.

Line 141: missing citation for the CERES_SYN1deg product.

**Results and Discussion**

Line 180: "Figure 1 shows a spectrum of OLR [...]". It is a directional radiance that the authors show on Fig. 1, not the OLR (as the authors define it before in the manuscript, the OLR is the radiance intensities integrated over all the angle directions).

Line 181: the authors write "The spectral ranges measured by IASI and FORUM are highlighted". For IASI, the spectral range extend to 2760 $cm^{-1}$ while Fig. 1 stops at 2250 $cm^{-1}$. Why stop at 2250? It should be mentioned in the text.

Same for line 185: "we focus here on [...] by IASI (645-2760 $cm^{-1}$) from 2006 onward" while in fact the authors limit to 2250 $cm^{-1}$ and focus on the period 2008-2016.

Line 189: when discussing the emission level, the authors could cite for example the paper of Whitburn et al. (2021) on the trends in spectrally resolved OLR.

Line 202: I agree that AW1 is more transparent than AW2, but to me this is not clearly visible on Fig. 2.

Fig. 2:
- It would be nice to add dashed lines to separate the different spectral regions on the two panels. This would help for the visualization.
- $N_2O$ is not visible on the bottom panel (I guess it is below $CH_4$). Also, $CH_4$ and Temperature should have different colors.

Line 260: Why are the limits set to 60° N/S. This should be justified.

Line 270: Mention this is a zonal average.

Line 282: If the bias in the $O_3$ band is due to the bias of the temperature of the lower atmospheric layers, I don't understand how it could be higher than the biases observed in the window regions?

Line 284: Why is the focus on tropical regions? I think it is important to justify it more clearly.

Line 285: Is it a weighted average on the Figure 4? I guess this is not so important because the focus is on tropical regions, but it might change slightly the results. If I understand correctly, Fig. 4 is simply an average of Fig. 3?

Line 290: Is the comparison with CERES based on a daily average OLR?

Lines 292-298: I agree with the explanation, but I think this paragraph should be moved elsewhere in the discussion.

Fig. 3: Maybe Fig. 3 and 4. could be merged into a two panel figures.

Line 307: What are the $CO_2$ concentrations used? (I mean, where have they been taken from).

Line 312: "[...] for the selected spectral channels of Table 2" → "averaged over the spectral bands of Table 2."

Fig. 5 and lines 313-319: To me what is especially interesting in fig. 5 is more that the bias seems rather constant over the year, except for panel D where a positive then a negative bias is observed with indeed an amplified seasonality. I think this change in the sign of the bias should be investigated.

Fig. 5: Panel C. Why this almost 1K bias is not visible on Fig. 3 and 4.

Lines 320-330: Here I think maps of $H_2O$ columns (from ERA5 for example) could greatly help for the interpretation of the results. Same for Ozone.

Fig. 7 (caption): Distribution of sea surface temperatures from model?

Line 356: ERA5 analysis: what latitude/longitude considered? Sea only? Why is the period considered not the same as in the rest of the paper (2008-2016).

**Technical corrections**

Line 52: Missing year in the ref to Harries et al.

Line 112: O3, CO2, N2O, […] Format the text to put all numbers in indices.

Line 124: 9.30 AM and PM.

Line 208: "in the wings of the […]"

---

## Author Comment (AC2)

**1 Reply to Referee 1**

We thank the referee for the very accurate feedback and comments, which helped to revise our work and will improve its presentation.

During the revision process, thanks to the first comment, we found an error in the procedure used to select the clear-sky nadir-looking IASI pixels. Although the error did not significantly affect the results of the study, Figures 3,5 and 6 of the paper will be updated as a result of this error correction. The updated figures are also contained in this document (Figure 8 at the end of the document and Figures 5, 6).

**2 Major Comments**

Below, the answers to the four major comments of the reviewer :

1. The selection of the clear-sky areas to analyse goes through a series of steps and eventually a set of clear sky spectra are obtained from the dataset of simulated radiances. Would it be possible to show a map with the regional density of pixels surviving the selection, to help identify which regions are most contributing to the analysis? For example, are the pixels used to create figure 3 distributed homogeneously or with a higher density somewhere? Or the analysis is robust enough to any regional bias?

   We agree with the reviewer that the spatial distribution of the measurements used to build the statistics is a useful information that may have an impact on the measured biases.

   The selection process of simulated and measured spectra that contribute to the presented statistics goes through the following steps.

[Figure]

Figure 1: The dots indicate the center of the ECE model cells for which a spectral radiance is simulated.

   **Simulated spectra**. To save computing time, ECE simulates spectra in correspondence of only once every 4 latitude x longitude grid cells. The dimension of model cells is 0.7×0.7°. The actual model cells for which spectra are simulated are shown in Figure 1. For each of these cells, we compute the monthly average radiance using only the simulated spectra with local solar time between 6 and 12 hours, only if the current cloud cover of the model cell is less than 30 %. We then compute the monthly zonal averages by averaging the monthly mean radiances relating to the model cells within the considered latitude belt. With this procedure all model cells contribute to the zonal mean with equal weight.

**Measured spectra**. IASI measured spectra are selected from 2°x2° cells centered on the ECE model cells for which spectra are simulated. On the one hand, the dimension of these cells is large enough to allow the selection of a sufficiently large number of IASI spectra. On the other hand, these cells do not overlap each other, thus each IASI measurement contributes only once to the statistics. For each of these cells, we compute the monthly average radiance using IASI measured spectra that meet the following conditions:

- The radiance is measured in day-time, in the near-nadir geometry, over the ocean, and corresponds to clear-sky conditions (cloud mask of AVHRR = 0).

- The measured radiance falls into a CERES grid cell, measured within 3 hours from the IASI observation time, with cloud cover less than 30 %. Since CERES grid cells have a dimension of 1x1 degree, similar to the ECE model cells, we apply the same threshold to the cloud cover in the CERES cells.

Finally, we compute the monthly zonal averages of observed radiances by averaging the monthly means obtained at the 2°x2° cells falling within the selected latitude belt.

The top and bottom panels of Fig. 2 show, respectively, the number of simulated and measured spectra that meet the above specified conditions, in the time interval from 2008 to 2016. The number of selected spectra is not homogeneously distributed across the globe. Most of the selected spectra are located in the subtropics ([15-30 N] and [15-30 S]), corresponding to the descending branch of the Hadley Cell. The pattern of the number of simulated and measured spectra that are selected, and thus contribute to cell- averages, is very similar in the tropical and sub-tropical regions, giving confidence on the fairness of the comparison method and on the main results of our work.

Note, however, that the filters used particularly affect the mid-latitudes ([45-60 N] and [45-60 S]), where only few IASI pixels survive to the selection process (see plots in Fig 3). The small number of observations meeting the mentioned criteria, together with the cloud cover bias described in Section 3.4 of the paper, could contribute to the bias found in the atmospheric window at these latitudes. This is one of the reasons why the intercomparisons presented in the paper focus on the tropical regions ([-30 S 30 N]), where we have a very large number of both modeled and observed spectra.

In the revised version of the paper we will include more comprehensive explanations on this issue. We also plan to include in a Supplement the plots presented in this reply.

[Figure]

(a)

[Figure]

(b)

Figure 2: Number of simulated (top) and observed (bottom) spectral radiances that contribute to the clear-sky statistics presented in the paper, for each lat x long cell.

[Figure]

Figure 3: Number of simulated (top) and observed (bottom) spectral radiances that contribute to the clear-sky zonal means presented in the paper.

2. Jacobians in Figure 2 are shown as normalized to the relative maximum. I understand that this is to show qualitatively the spectral sensitivity of the radiance to various parameters, but would it make sense to also show the absolute values somehow (for example in Figure 6)? It would be interesting for example to estimate if the biases in temperature and water vapour concentration shown in figure 8 would indeed reasonably map into the observed radiance/brightness temperature biases discussed for figure 6 and figure 5c

We agree that this information will be useful in the discussion of specific spectral biases.

In this regard, panels A and B of Fig.4 show the absolute values of the jacobians of temperature (A) and water vapour (B). This plot will be added to the supplementary material in the revised manuscript.

[Figure]

Figure 4: Jacobians of Temperature and Water Vapour

In addition, we will update Figures 5 and 6 of the paper with the new Figures 5 and 6 of this document. These new plots represent the BT of the climate model and IASI in different spectral bands and the absolute values of the jacobians of the temperature and water vapour concentration, in order to highlight the layers of the atmosphere that affect the most the

TOA radiance.

[Figure]

Figure 5: Brightness Temperature (BT) averaged in different spectral intervals and absolute values of temperature jacobians in the respectively spectral bands. This is the new figure 5 of the paper

[Figure]

Figure 6: In the panels on the left, the absolute values of water vapour and temperature jacobians at 1400 and 730 cm$^{-1}$. On the right, the Brightness Temperature (BT) of IASI and ECE at 1400 cm$^{-1}$. This is the new figure 6 of the paper

In all these plots, to simplify the spectral channels description, we shifted the 10 cm$^{-1}$ spectral bands used. When we indicate a specific wavenumber, we refer to the value of the spectral radiance convolved/averaged in the $\pm 5$ cm$^{-1}$ interval centered about the given wavenumber. For example, when we indicate "700 cm$^{-1}$", we refer to the spectral radiance convolved/averaged in the range from 695 to 705 cm$^{-1}$. The presented jacobians are represented in the same intervals.

As suggested, we also provide an estimation of the BT bias inferred from the temperature and water vapour biases obtained from the comparison between the EC-Earth model and ERA5 (Figure 8 of the paper).

First, we computed the mean profile of these temperature and water vapour biases along the

tropical latitudes [-30S, 30N]. Then, we performed the scalar product of these profiles with the respective Jacobians, computed from the profiles of a standard tropical atmosphere.

The result of this estimation is summarized in Table 1.

| Spectral Channel( cm$^{-1}$) | 660 | 700 | 730 | 850 | 1400 |
|---|---|---|---|---|---|
| Most sensible region to temperature (km) | [25-45] | [5-15] and [20-35] | [3-10] | [0-5] | [3-10] |
| Most sensible region to water vapour (km) | – | – | – | – | [5-20] |
| BT bias - ERA5 (K) | +2.5 | +0.8 | -0.2 | -0.2 | +0.7 |
| BT bias - IASI (K) | +3.5 | +1.5 | -0.8 | -0.2 | +0.5 |

Table 1: Inferred BT biases estimated from the comparison between climate model outputs and ERA5 data and from the comparison of climate model and IASI BT climatologies

The result of our analysis provides a stronger positive bias in the stratosphere. This difference also causes a slightly higher BT bias at 700 cm$^{-1}$.

In addition, we also have a more pronounced negative bias at 730 cm$^{-1}$, most likely produced by a higher negative temperature bias of the model in the middle troposphere. Finally, the BT biases for the spectral bands at 850 and 1400 cm$^{-1}$ show a very good agreement with the result obtained from the reanalysis.

In both the estimations, the positive BT bias at 1400 cm$^{-1}$ is caused by the negative bias of water vapour concentration in the upper troposphere of the model.

We agree that this result can be interesting for the paper and we will add it to the revised version of the manuscript.

3. Figure 5 panel B shows that EC-Earth has a positive temperature bias in the region 5-15km in the 30S- 30N area as seen in the 700-710 cm-1 spectral region. Yet in the comparison against ERA5 that area shows generally a negative bias up to 15km (positive biases are seen north and south of 30 degrees, above the tropopause). This seems in contradiction to what it is said at line 360 Ch. 3.4

In the spectral band centered at 700 cm$^{-1}$ we find a positive bias of the BT model with respect to BT measured by IASI. This result seems not to be in agreement with the negative bias highlighted in the comparison with the ERA5 reanalysis.

However, as explained in line 360 and how visible in Panel B of Fig. 5, it is evident that this spectral region is also affected by stratospheric temperatures, which are positively biased in the model.

As a result, the strong negative bias at the tropopause (Figure 8), is masked by the positive bias of the stratosphere. In fact, also the BT bias, estimated from the profile of temperature bias of the reanalysis at 700 cm$^{-1}$, is positive (see Table 1).

We will stress this aspect in the revised manuscript adding the results of the previous estimation, presented in table 1, and adding the new Figure 5, where the profiles of the jacobians are shown.

4. In the conclusion I'd like to see a clearer discussion of the limitations of the presented method, namely the difficulty of making sure that a meaningful comparison between simulated and measured radiances can be obtained (this is briefly touched in chapter 3.4 regarding sampling biases). The analysis presented in this work does indeed show the advantage of having spectrally resolved OLR computed from the model fields overcoming some of the limitations of retrieval products, but, on the other hand, assumptions and parameterisations are still needed to perform accurate radiative transfer computations. Restricting the analysis to clear-sky ocean areas circumvents the difficulties of a correct simulation of surface radiative properties or the representation of cloud microphysical characteristics whose uncertainties might create larger differences than the model biases one attempts to investigate. Can these shortcomings be somehow mitigated, or is the method really only suited for a limited set of cases?

Thanks for this comment, which deserves some discussion in the paper. In this particular case, we have been working within a simplified framework considering only clear-sky spectra measured and simulated over the Tropical Ocean. However, as shown, the study of the clear sky also involves different problems with the average data distribution. In our case, filters applied to ensure the consistency of the comparisons between model and observation, significantly reduce the number of selected spectra at high latitudes. As a result, reliable results are obtained only limited to the Tropical Oceans.

In contrast, in the all sky analysis we do not have the issues of not homogeneous sampling. In this case, the major problem involves the treatment of clouds in the climate model. Climate models have horizontal grid spacing of tens of kilometers and cloud variability inside a grid cell has to be treated statistically. One possible way consists in decomposing the cell of the model in a set of subcolumns, whose spatial dimension is comparable with the field of view of the instruments. Thus, the radiative computation can be performed over the single subcolumns.

In both cases, to have reliable results, it is necessary to start working under the simplest conditions (clear-sky, ocean, etc.) Once the biases have been investigated for the simplest cases, it can be possible to gradually work in more general conditions.

This can be possible if we exploit and compare the results from other existing procedures (reanalysis, comparison of Level 2 products of instruments).

**3  Minor Comments**

1. Sect. 1, line 39: "and, thus, to a specific parameter": what is it meant here by parameter? An active model variable such as temperature or concentration of a particular atmospheric constituent or rather a model parameter governing a particular process/parametrisation in the model? I think that it is difficult that a single model parameter, if the latter is intended, would be able to explain a bias in a particular spectral region. Indeed, many of the parameters tuned in EC-Earth to achieve satisfactory radiative fluxes at TOA, affect cloud parametrisations. Please, clarify

   We refer to an active model variable (temperature, concentration of gases, etc.).

2. Sect. 1, line 43: "1970s, but only starting from the 2000s long term. . . ." OK, we will correct it in the new version.

3. Sect. 1, line 56: "In anticipation of FORUM measurements. . . .". This is repeated few time in the document: does it mean here something along the lines of "To demonstrate the potential of the future FORUM measurements"? OK, we will correct it in the new version.

4. Sect. 1, line 65: "MIR" has not been defined while FIR yes. Ok, we will correct it in the new version.

5. Sect 1, line 79: "paves the way for direct assimilation" Not clear here: why the implementation of an online simulator would help in that direction?

   Here we mean that the assimilation process in a model is generally performed trough the assimilation of the radiances. Therefore, the possibility to provide radiances in a climate model, in this case through an online simulator, could help in this direction. However, to avoid confusion, we will remove this sentence from the manuscript because it needs a more accurate explanation.

6. Sect. 2.1.1, line 101: CLOUDSAT is an active radar. ISCCP is not a sensor if it means the International Satellite Cloud Climatology Project. References and full description of the acronym would help here, as done for RTTOV mentioned later, for example. OK, we will correct it in the new version.

7. Sec 2.2.1, line 137: is there a reference for the AVHRR cloud mask? Thanks, we will add the reference.

8. Sec 2.2.2: I think there should be references to the CERES dataset used Thanks, we will add the reference.

9. Sec 3.1, line 197: "partial derivatives of radiance with respect to the most relevant atmospheric parameter". Perhaps a more general definition like "with respect to any relevant atmospheric variable. Here we show the most relevant..." OK, we will correct it in the revised manuscript.

10. Sec 3.3, line 298: The example discussed in this section shows that spectral OLR are indeed useful to provide a complete view of model biases, but it also shows the difficulty to select meaningful area to compare with measurement, thus limiting somehow the possibility to map directly a bias to a parameter in a global model. Perhaps the sentence can be rephrased to reflect this limitation.

Ok, we will stress this aspect.

11. Sec 3.3, line 330: "witnessed by the negative sign of the water Jacobian" -¿ "as represented by the negative values of the water vapour Jacobian"

Ok, thank you for the suggestion.

12. Sec 4, line 392: The results are only shown for ocean areas, land biases are not really discussed in depth in the text. Is it worth it reporting it in the conclusions?

Ok, we agree with you. Thereofre, we will remove the sentences related to the biases over lands since we did not focus on in the paper.

13. The representation of the differences in specific humidity as percentage tends to highlight the stratosphere due to the very low concentration of water vapour there. The absolute differences could perhaps be better related to the absolute value of the Jacobian that could be shown e.g. in Figure 6?

Figures 7 describe the absolute difference of water vapour concentration between the reanalysis and the ECE model. More information about the possibility to link these differences to the absolute jacobians values are contained in the answer to question 2 of the major comments.

We will add these figures in the Supplementary material of the revised paper.

[Figure]

Figure 7: Absolute difference of humidity concentration between model and ERA5 data in the lower troposphere (a) and in the upper troposphere-lower stratosphere (b)

[Figure]

Figure 8: BT difference (model-IASI). New figure 3 of the paper

---

## Author Comment (AC3)

**1 Reply to Referee 2**

We thank the referee for the very accurate feedback and comments, which helped revise our work and will improve its presentation.

During the revision process, we found an error in the procedure used to select the clear-sky nadir-looking IASI pixels. Although the error did not significantly affect the results of the study, Figures 3,5 and 6 of the paper will be updated as a result of this error correction. The updated figures are also contained in this document (Figures 4, 14, 15).

1. **Subsection 2.1.1** Could the authors describe a bit more how the EC Earth model works (maybe accompanied by a conceptual flowchart)? In particular, I am not sure to understand from where are all the prescribed parameters coming from. To me, it should be explained a bit more in the manuscript. Or at least, the authors should mention clearly in what paper an extensive description of the model can be found.

    **3.2 Model comparison strategy**

    I think this section is not clear enough and should be partly rewritten.

    For instance, "The model was run with prescribed SSTs and SIE, [. . .]" It is not clear to me where are these information coming from (see my previous comment)

    EC-Earth3 is a coupled climate model in which the atmospheric model IFS (cy36r1), which includes the representation of land processes (HTESSEL, Balsamo et al., 2009), is coupled to the NEMO ocean model, including LIM3 as sea-ice component. The EC-Earth model is thoroughly described in Döscher et al. (2022), which we refer to for a detailed model description. The IFS is a general circulation model which solves the atmospheric dynamics from primitive equations and includes parametrizations representing physical processes (radiation, microphysics, ..) and non-resolved processes (convection, turbulence, ..). In this work, we used the EC-Earth model in an atmosphere-only configuration, this means that the atmospheric model IFS has been run without the oceanic counterpart and with imposed boundary conditions at sea surface. This configuration is the one used for the Atmospheric Model Intercomparison Project (AMIP) simulations in the context of CMIP6. The prescribed SST and SIC come from the AMIP protocol configuration for CMIP6 (Eyring et al., 2016) and are provided as standard input to all models participating to CMIP6 (see also https://pcmdi.llnl.gov/mips/amip/ and https://esgf-node.llnl.gov/projects/input4mips/). The dataset is created with the procedure described in Hurrel et al. (2008) and merges the HadISST observational dataset (since 1870) to the more recent NOAA-OI (since 1981). EC-Earth reads the SST and SIC as mid-month boundary conditions, which are then interpolated daily in the model run. We will add more details on this in the revised manuscript.

    Hurrell, J. W., J. J. Hack, D. Shea, J. M. Caron, and J. Rosinski, 2008: A New Sea Surface Temperature and Sea Ice Boundary Dataset for the Community Atmosphere Model. Journal of Climate, 21, 5145–5153, https://doi.org/10.1175/2008jcli2292.1

2. If I understood well, the authors built monthly climatology of IASI radiance (near-nadir). Only cloud free pixels are considered for the averaging. What is the grid size of these monthly climatology? 1°x1°? Then, for the comparison with the model, from these monthly climatology, the authors only consider the grid cells that show a monthly mean cloud cover (according to CERES) lower than 30 %. Is that right?

    The selection process of simulated and measured spectra that contribute to the presented statistics goes through the following steps.

    **Simulated spectra**. To save computing time, ECE simulates spectra in correspondence of only once every 4 latitude x longitude grid cells. The dimension of model cells is $0.7 \times 0.7°$. The actual model cells for which spectra are simulated are shown in Figure 1. For each of these cells, we compute the monthly average radiance using only the simulated spectra with local solar time between 6 and 12 hours, only if the current cloud cover of the model cell is less than 30 %. We then compute the monthly zonal averages by averaging the monthly mean radiances relating to the model cells within the considered latitude belt. With this procedure all model cells contribute to the zonal mean with equal weight.

[Figure]

Figure 1: The dots indicate the center of the ECE model cells for which a spectral radiance is simulated.

**Measured spectra**. IASI measured spectra are selected from 2°x2° cells centered on the ECE model cells for which spectra are simulated. On the one hand, the dimension of these cells is large enough to allow the selection of a sufficiently large number of IASI spectra. On the other hand, these cells do not overlap each other, thus each IASI measurement contributes only once to the statistics. For each of these cells, we compute the monthly average radiance using IASI measured spectra that meet the following conditions:

- The radiance is measured in day-time, in the near-nadir geometry, over the ocean, and corresponds to clear-sky conditions (cloud mask of AVHRR = 0).
- The measured radiance falls into a CERES grid cell, measured within 3 hours from the IASI observation time, with cloud cover less than 30 %. Since CERES grid cells have a dimension of 1x1 degree, similar to the ECE model cells, applying the same threshold to the cloud cover we ensure consistency of the atmospheric conditions between model and observations.

Finally, we compute the monthly zonal averages of observed radiances by averaging the monthly means obtained at the 2°x2° cells falling within the selected latitude belt.

The top and bottom panels of Fig. 2 show, respectively, the number of simulated and measured spectra that meet the above specified conditions, in the time interval from 2008 to 2016. The number of selected spectra is not homogeneously distributed across the globe. Most of the selected spectra are located in the subtropics ([15-30 N] and [15-30 S]), corresponding to the descending branch of the Hadley Cell. The pattern of the number of selected spectra is very similar in simulations and measurements in the tropical and sub-tropical regions, giving confidence on the fairness of the comparison method and on the main results of our work.

Note, however, that the filters used particularly affect the mid-latitudes ([45-60 N] and [45-60 S]), where only few IASI pixels survive to the selection process (see plots in Fig 3). The small number of observations meeting the mentioned criteria, together with the cloud cover bias described in Section 3.4 of the paper, could contribute to the bias found in the atmospheric window at these latitudes. This is one of the reasons why we mostly focus on the tropical regions ([-30 S 30 N]), where we have a large number of both modeled and observed spectra.

In the revised version of the paper we will include more details on this issue. We also plan to include the plots presented in this reply in the Supplementary material.

[Figure]

(a)

[Figure]

(b)

Figure 2: Number of simulated (top) and observed (bottom) spectral radiances that contribute to the clear-sky statistics presented in the paper, for each lat x long cell.

[Figure]

Figure 3: Number of simulated (top) and observed (bottom) spectral radiances that contribute to the clear-sky zonal means presented in the paper.

3. How was this 30% threshold chosen? Why not 40% or 20% for example? I think it should be better justified.

As explained in the paper, this filter was applied with the aim to mitigate the potential bias due to the way the radiative computations are performed in the model in clear-sky condition. In fact, the radiative transfer computation in clear-sky condition in climate models exploits the same all-sky properties (profiles of temperature, humidity, etc.) but with clouds removed. This could produce a negative bias in the synthetic radiances when compared to clear-sky observations.

The choice of the 30% threshold is the result of a trade-off between reducing the impact of this potential source of bias and keeping a significant number of measurements in the analysis. In principle, a lower threshold would be more desirable, but at the same time this reduces the statistics. We performed some tests and we judged that the use of a 10% or 20% threshold was too strict because the dataset was significantly reduced. On the other end, the impact of the clear-sky computation bias with the 30% threshold appeared to be quite small, as demonstrated a-posteriori by the good match in the atmospheric window at tropics.

We will extend the discussion on this point in the new version of the manuscript, also providing some quantitative estimates on the impact of this threshold on the number of observations available.

4. A map showing the grid cells that are kept for the comparison would be nice

The map is shown in figure 1.

5. Also, in the IASI monthly mean radiances, how many measurements are present on average per month in each grid cell? If the number is too low (which might be the case for some of the pixels as only near-nadir observations are considered), the comparison with the model might be biased as a few number of observations is not representative of a whole month. This should be kept in mind for the analysis. It might be good to add a postfiltering to remove from the comparison the grid cells that contain too few observations

The following table describes the statistics of the clear-sky spectra used in the climatology for IASI and EC-Earth(Table 1). For each latitudinal band, the tables show the total number of grid cells (second column), the number of clear sky grid cells with at least 1 measured or simulated spectrum per month (third column, fourth columns) and the average number of spectra per month in these grid cells (fifth and sixth columns).

| Lat Band | Tot gcells | N. of gcells (IASI) | N. of gcells (ECE) | N. of obs. (IASI) | N. of obs. (ECE) |
|---|---|---|---|---|---|
| $60S - 45S$ | 493 | 33 | 428 | 8 | 3 |
| $45S - 30S$ | 506 | 147 | 495 | 9 | 5 |
| $30S - 15S$ | 573 | 332 | 565 | 14 | 10 |
| $15S - 0S$ | 497 | 317 | 457 | 17 | 10 |
| $0N - 15N$ | 501 | 256 | 447 | 15 | 8 |
| $15N - 30N$ | 491 | 320 | 477 | 18 | 10 |
| $30N - 45N$ | 304 | 116 | 291 | 15 | 7 |
| $45N - 60N$ | 221 | 34 | 198 | 9 | 4 |

Table 1: IASI and ECE point statistics

In general, for IASI only a few grid cells per month have spectra at the highest latitudes. In contrast, simulated spectra can be detected at almost all grid cells in a month. This difference can affect the comparison, especially at high and mid-latitudes, where the large difference in the number of available cells may represent a source of bias. On the other side, in the tropical oceans this difference is smaller and the statistics tends to be similar.

To check the sensitivity of our analysis in this respect, we applied a threshold for the minimum number of spectra in each grid cell, as the reviewer suggests.

More in detail, we removed from the comparison the grid cells containing less than 5 spectra per month. The statistics of the clear-sky spectra under this condition is shown in tables 2.

| Lat Band | Tot gcells | N. of gcells (IASI) | N. of gcells (ECE) | N. of obs. (IASI) | N. of obs. (ECE) |
|---|---|---|---|---|---|
| $60S - 45S$ | 493 | 15 | 41 | 16 | 7 |
| $45S - 30S$ | 506 | 72 | 206 | 15 | 8 |
| $30S - 15S$ | 573 | 220 | 439 | 19 | 12 |
| $15S - 0S$ | 497 | 239 | 321 | 24 | 15 |
| $0N - 15N$ | 501 | 171 | 247 | 22 | 13 |
| $15N - 30N$ | 491 | 239 | 356 | 23 | 13 |
| $30N - 45N$ | 304 | 74 | 139 | 25 | 11 |
| $45N - 60N$ | 221 | 18 | 41 | 16 | 8 |

Table 2: Statistics of IASI spectra after the application of the threshold

The number of cells with at least 5 spectra per month (third column of the tables) is strongly reduced, in particular in the model, since here we have an average of few spectra per month over most grid points.

After the application of the threshold, the pattern of the BT biases is the same, with larger differences occurring at high latitudes and over tropical latitudes in the water vapour band. Both the plots are shown in Figures 4 (without the threshold) and 5 (with the threshold).

[Figure]

Figure 4: BT difference (model-IASI) without the application of the threshold

6. Lines 252-253: "In fact, monthly means of observed and simulated outgoing radiances are computed over the same spatial grid by associating IASI measurements to the nearest EC-Earth grid point.". I am not sure I got the idea here. If I understood well, for the time correspondence between the model simulations and the IASI observations, only the model output of the 6-12 AM and PM time are considered? Both are then averaged to obtain a "daily" mean (and same for IASI, ascending and descending nodes are averaged to get a daily mean?)? And this is the reason why you only considered sea measurements for the analysis (because of the very high diurnal cycle of the radiance over land)?

In this work we compared the simulated spectra of ECE with local solar time between 6 and 12 hour to the observed spectra of the descending node of IASI (ground track at 9:30 AM at the equator).

On the basis of the answer given to the question 2, we will modify the lines 252 - 257 in the revised manuscript .We will also highlight that the comparison is limited to the day-time.

As pointed out by the reviewer, this procedure works well over the ocean but it is not reliable over land. Here, simulated spectra at closer timesteps (e.g. 3 hours or less) would be needed to represent the large diurnal cycle.

[Figure]

Figure 5: BT difference (model-IASI) after the application of the threshold

**3.3 Assessment of EC-Earth biases in simulated clear-sky radiances with respect to IASI measurements**

The descriptions in this section should be more quantitative. For example, line 274 (but also elsewhere): "[...] a small positive bias". Of how much?

Thanks for the suggestion. The new version of the manuscript will be more quantitative in the description of the different biases. As discussed in the answer to question 28, we will also add a table with the estimation of the biases computed from our spectral analysis and that inferred from the temperature and water vapour biases obtained from the comparison with the reanalysis.

7. Are the biases the same every year or is there differences from year to year? Maybe a time series showing the differences between the model and IASI (averaged on the different bands WV1, WV2, ... for example) would be nice. I am also wondering if there is a difference in the biases between day and night

In general, we do not have significant variations of the BT difference during the 9 years.

Figure 6 shows the time series of the BT in four spectral bands (CO2 core band, atmospheric window AW1, ozone band and water vapour band WV2). For all these spectral intervals, despite the variations in BT, the difference between model and observations are quite constant over time.

In this work we limited the study to the day-time spectra and we cannot quickly produce a statistics for the night. However, we agree it is an interesting analysis and we would like to analyze it in the continuation of this work.

8. Another plot that could greatly help to interpret the results and could be a nice addition to the paper are global maps of the difference between IASI and the model for some selected channels (instead of a zonal average of the difference) (as done in Whitburn et al. 2021 for example). In particular, this would allow us to highlight potential compensating biases between different regions

The following figures (7, 8, 9, 10, 11) will be added to the supplementary material of the manuscript. The global BT difference is shown for channels 660 $cm^{-1}$, 700 $cm^{-1}$, 730 $cm^{-1}$, 850 $cm^{-1}$, 1400 $cm^{-1}$.

We agree these plots are particularly useful for verifying the presence of spatial compensation errors. However, they are reliable only over the Tropical Ocean (between 30 S and 30 N), where we have a complete time series of IASI and ECE monthly averaged BT for almost all the grid cells.

[Figure]

Figure 6: Time series in different spectral bands of the BT measured and simulated over the Tropical Ocean [-30 S, 30 N] from 2009 to 2016

Over these latitutudes, we find some compensating biases only at 850 cm$^{-1}$, in the atmospheric window. However, these differences are generally small, within 1 K.

Finally, the positive bias at 1400 cm$^{-1}$ is particularly evident along the tropical latitudes, from -15 S to +15 N. This result could be interesting for a more accurate discussion about the deficiency of water vapour concentration in the upper troposphere of the model highlighted in the paper and shown in Fig. 15.

As suggested, we will add the reference to the interesting paper of Whitburn.

[Figure]

Figure 7: BT difference (model - IASI) at 660 $\text{cm}^{-1}$

[Figure]

Figure 8: BT difference (model - IASI) at 700 $\text{cm}^{-1}$

[Figure]

Figure 9: BT difference (model - IASI) at 730 cm$^{-1}$

[Figure]

Figure 10: BT difference (model - IASI) at 850 cm$^{-1}$

[Figure]

Figure 11: BT difference (model - IASI) at 1400 cm$^{-1}$

[Figure]

Figure 12: Figure 3 of the paper expressed in radiance

9. It would be nice to see Fig. 3 in W m-2 to evaluate more easily the impact of the differences (maybe in appendix)

In Figure we represent the Figure 3 of the paper expressed in radiance. We think that for this kind of plot the BT is clearer than the radiance because it can highlight the differences over the whole spectral range. On the contrary, the radiance difference in the Plankian tail becomes too small to be visible.

**SPECIFIC COMMENTS**
**ABSTRACT**

1. Line 15: If I am not mistaken, I think it is "Metop" and "MetOp". To be checked and replace everywhere.

Ok, we found both the acronyms are used but we will substitute MetOp with Metop and we will be coherent in all the revised manuscript.

2. Line 18-19: "[...] while a cold bias occurs over land". The authors didn't mention that they were talking about sea just before. Please clarify.

Ok

**INTRODUCTION**

3. Line 41: Missing references for AIRS and IASI.

We will add the references in the new version

4. Line 46: Also mention the existence of a spectrally resolved OLR product at 10 cm-1 derived from AIRS measurements (e.g. Huang et al. 2008, Chen et al. 2016).

Ok, thanks for the suggestion

5. Appendix A: I don't really see the added value of Appendix A for the paper. The information presented are very general and the different formula can be found in many atmospheric science books and published papers.

It is surely true, but we think that these concepts are less common among the climate modelling community, especially when referring to the spectral dimension.

**DATA AND METHODS**

6. Line 91: Please defined IFS.

Ok

7. Line 132: FCDR: missing citation. Also, mention that this dataset is reprocessed with the latest version of the L1C and is consistent over the whole time series. I guess this is why you chose to use it?

Yes, correct. We will add the citation and specify why we use this dataset.

8. Line 136: [. . . ] the 8 pixels closest to the nadir view. Thus, the clear-sky [. . . ]. There is a problem with the transition between the two sentences here.

Ok, we will correct them.

9. Line 137: AVHRR cloud dataset: cite Guidard et al. (2011). Is there a reason why this cloud product was chosen over the L2 cloud product? For consistency over the whole time series? Maybe it would be nice to justify it.

Yes, thanks for the suggested reference. We use all data contained in the reprocessed L1 data, including the information about the cloud cover provided by the AVHRR.

10. Line 141: missing citation for the CERESSYN1deg product.

Ok, we will add it in the new version.

**RESULTS AND DISCUSSION**

11. Line 180: "Figure 1 shows a spectrum of OLR [. . . ]". It is a directional radiance that the authors show on Fig. 1, not the OLR (as the authors define it before in the manuscript, the OLR is the radiance intensities integrated over all the angle directions).

Ok, we agree with you and we will correct this sentence.

12. Line 181: the authors write "The spectral ranges measured by IASI and FORUM are highlighted". For IASI, the spectral range extend to 2760 cm-1 while Fig. 1 stops at 2250 cm-1. Why stop at 2250? It should be mentioned in the text.

We focused on the spectral range containing all the spectral intervals exploited for the detection of the model biases. This will be specified in the text.

13. Same for line 185: "we focus here on [. . . ] by IASI (645-2760 cm-1) from 2006 onward" while in fact the authors limit to 2250 cm-1 and focus on the period 2008-2016.

Ok, thanks for this correction. We will include it in the revised paper.

14. Line 189: when discussing the emission level, the authors could cite for example the paper of Whitburn et al. (2021) on the trends in spectrally resolved OLR.

Yes, we will add the reference to the new paper of Whitburn.

15. Line 202: I agree that AW1 is more transparent than AW2, but to me this is not clearly visible on Fig. 2.

Yes, we agree. However, in this kind of plot we used it is very hard to highlight the light lines of the water vapour jacobian.

16. Fig. 2: - It would be nice to add dashed lines to separate the different spectral regions on the two panels. This would help in visualization. - N2O is not visible on the bottom panel (I guess it is below CH4). Also, CH4 and temperature should have different colors.

The new plot is shown in Figure 13. The N2O is completely covered by methane, so we removed it. Accordingly, Figure 2 of the paper will be replaced.

17. Line 260: Why are the limits set to 60° N/S. This should be justified.

Over these latitudes the number of spectra, especially measured spectra, is too low to perform the comparison. We will justify it in the new version of the manuscript.

18. Line 270: Mention this is a zonal average

Ok.

[Figure]

Figure 13: Absolute values of normalized Jacobians computed for a tropical standard atmosphere with respect to the temperature (top panel) and gases concentration (bottom panel).

19. Line 282: If the bias in the O3 band is due to the bias of the temperature of the lower atmospheric layers, I don't understand how it could be higher than the biases observed in the window regions?

Thanks for the comment. As explained in line 209, the 03 band is affected by surface, lower troposphere and stratospheric temperature. So, the temperature bias of the lower atmospheric layers only contributes to the overall bias we detect in this spectral interval. We will correct the sentence in the revised paper to avoid confusion.

20. Line 284: Why is the focus on tropical regions? I think it is important to justify it more clearly.

The filters used to select the spectra particularly affect the mid-latitudes ([45-60 N] and [45-60 S]), where only few IASI pixels survive to the selection process (see plots in Fig 3). The small number of observations meeting the mentioned criteria, together with the cloud cover bias described in Section 3.4 of the paper, could contribute to the bias found in the atmospheric window at these latitudes. This motivation will be addressed in the paper.

21. Line 285: Is it a weighted average on the Figure 4? I guess this is not so important because the focus is on tropical regions, but it might change slightly the results. If I understand correctly, Fig. 4 is simply an average of Fig. 3?

Yes, it is a weighted average where the weight is the cosine of the latitude. Yes, it is right.

22. Line 290: Is the comparison with CERES based on a daily average OLR?

The answer to this question is contained in the reply to the question 2 of this document.

23. Lines 292-298: I agree with the explanation, but I think this paragraph should be moved elsewhere in the discussion.

Ok, thank you for the suggestion.

24. Fig. 3: Maybe Fig. 3 and 4. could be merged into a two panel figures.

We tried to merge the pictures into a two panels figure but the result is not so good.

25. Line 307: What are the CO2 concentrations used? (I mean, where have they been taken from).

The CO2 concentrations used are the global and annual mean observed values (interpolated daily from one year to the other) provided by CMIP6 for the AMIP simulations and referenced in Meinshausen et al. (2017). For the last 2 years (2015-16), strictly speaking we adopted the SSP2-4.5 scenario data (Meinshausen et al., 2020), which however matches observations until 2017. We will add more details on this in the text.

Meinshausen, M., Vogel, E., Nauels, A., Lorbacher, K., Meinshausen, N., Etheridge, D.M., Fraser, P.J., Montzka, S.A., Rayner, P.J., Trudinger, C.M., Krummel, P.B., Beyerle, U., Canadell, J.G., Daniel, J.S., Enting, I.G., Law, R.M., Lunder, C.R., O'Doherty, S., Prinn, R.G., Reimann, S., Rubino, M., Velders, G.J.M., Vollmer, M.K., Wang, R.H.J., Weiss, R., 2017. Historical greenhouse gas concentrations for climate modelling (CMIP6). Geoscientific Model Development 10, 2057–2116. https://doi.org/10.5194/gmd-10-2057-2017

Meinshausen, M., Nicholls, Z.R.J., Lewis, J., Gidden, M.J., Vogel, E., Freund, M., Beyerle, U., Gessner, C., Nauels, A., Bauer, N., Canadell, J.G., Daniel, J.S., John, A., Krummel, P.B., Luderer, G., Meinshausen, N., Montzka, S.A., Rayner, P.J., Reimann, S., Smith, S.J., van den Berg, M., Velders, G.J.M., Vollmer, M.K., Wang, R.H.J., 2020. The shared socio-economic pathway (SSP) greenhouse gas concentrations and their extensions to 2500. Geoscientific Model Development 13, 3571–3605. https://doi.org/10.5194/gmd-13-3571-2020

26. Line 312: "[...] for the selected spectral channels of Table 2" → "averaged over the spectral bands of Table 2."

Ok, we will correct it in the manuscript.

27. Fig. 5 and lines 313-319: To me what is especially interesting in fig. 5 is more that the bias seems rather constant over the year, except for panel D where a positive then a negative bias is observed with indeed an amplified seasonality. I think this change in the sign of the bias should be investigated.

The change in the sign of the bias in panel D has been removed after the correction of the error affecting the selection of the clear-sky IASI points, as mentioned at the beginning of this document. The new figure is shown in Fig. 14 - Panel D. It remains a more peaked seasonality in the ECE curve, which is however within the standard deviation of the two curves.

[Figure]

Figure 14: Brightness Temperature (BT) averaged in different spectral intervals and absolute values of temperature jacobians in the respectively spectral bands. This is the new figure 5 of the paper

Fig. 5: Panel C. Why this almost 1K bias is not visible on Fig. 3 and 4.

This bias can be noticed in the Figure 4 even if it is the only negative value in all the CO2 band. For this reason, it corresponds to a very thin blue line in Figure 3. Maybe, it can be better appreciated from Figure 12.

28. Lines 320-330: Here I think maps of H2O columns (from ERA5 for example) could greatly help for the interpretation of the results. Same for Ozone.

Yes, we agree.

Figures 16 describe the absolute difference of water vapour concentration between the reanalysis and the ECE model.

We will add these plots to the new version of the manuscript. In addition, we will add an estimation of the BT biases inferred starting from the differences of temperature and humidity found from the comparison with ERA5 and the computation of the respective jacobians (Figure 14 and 15).

29. Fig. 7 (caption): Distribution of sea surface temperatures from model?

Yes, correct.

[Figure]

Figure 15: In the panels on the left, the absolute values of water vapour and temperature jacobians at 1400 and 730 cm$^{-1}$. On the right, the Brightness Temperature (BT) of IASI and ECE at 1400 cm$^{-1}$. This is the new figure 6 of the paper

30. Line 356: ERA5 analysis: what latitude/longitude considered? Sea only? Why is the period considered not the same as in the rest of the paper (2008-2016).

Here we considered the differences (model - era5) averaged over all the latitude (from 90 S to 90N) over Ocean. No significant differences arise when we reduce the number of years to the period 2008-2016.

[Figure]

Figure 16: Absolute difference of humidity concentration between model and ERA5 data

31. **TECHNICAL CORRECTIONS**

32. Line 52: Missing year in the ref to Harries et al.

     Ok

33. Line 112: O3, CO2, N2O, [. . . ] Format the text to put all numbers in indices.

     Ok

34. Line 124: 9.30 AM and PM.

     Ok

35. Line 208: "in the wings of the [. . . ]"

     Ok